# Cross-Image Context for Single Image Inpainting

**Tingliang Feng, Wei Feng, Weiqi Li, Di Lin**[*]
College of Intelligence and Computing, Tianjin University
{fengtl, wicky}@tju.edu.cn, wfeng@ieee.org, Ande.lin1988@gmail.com

## Abstract

Visual context is of crucial importance for image inpainting. The contextual information captures the appearance and semantic correlation between the image regions, helping to propagate the information of the complete regions for reasoning the content of the corrupted regions. Many inpainting methods compute the visual context based on the regions within a single image. In this paper, we propose the *Cross-Image Context Memory* (CICM) for learning and using the cross-image context to recover the corrupted regions. CICM consists of multiple sets of the cross-image features learned from the image regions with different visual patterns. The regional features are learned across different images, thus providing richer context that benefits the inpainting task. The experimental results demonstrate the effectiveness and generalization of CICM, which achieves state-of-the-art performances on various datasets for single image inpainting.

## 1 Introduction

Image inpainting requires understanding the object appearances and semantic categories in the images. The recent success of image inpainting lies in the end-to-end training of the deep networks, which are good at learning the appearance and semantic features from large-scale data.

Generally, the inpainting networks [1, 2, 3, 4, 5, 6, 7, 8] yield the regional features that capture the appearance and semantic correlation between the image regions, thus providing the visual context. Based on the visual context, the relevant information of the complete regions can be propagated to the corrupted regions, whose contents are consequently reasoned. Conventionally, the context propagation takes place between the regions within the isolated image. However, the existing methods, which heavily rely on the visual context, face a dilemma, where the corrupted regions naturally lack the visual information for computing the reliable context. It lets the learned context lose the critical relationships between the image regions, eventually degrading the inpainting performance.

In this paper, we advocate the idea of learning the cross-image context to recover the corrupted image regions. Alongside the inpainting network, we construct an external memory, where we store the regional features of the complete regions across different images. For inpainting, we search CICM to achieve the regional features, which are potentially relevant to the corrupted regions. Rather than depending on the regional features learned from a single image, we employ external memory to provide richer cross-image context. It allows the regional information from different images to be propagated to the corrupted regions in an image, where the lost contents are better restored.

Specifically, we propose the external memory, *Cross-Image Context Memory* (CICM), which can be equipped with popular inpainting networks. CICM consists of multiple feature sets. Each set contains the cross-image features. *Context Generalization* injects the regional features, which are learned from different images, into the cross-image features. We let the cross-image features in different sets have diversity. Thus, we allow different sets capture a richer cross-image context. In the same set, we

---

[*]Di Lin is the corresponding author of this paper.

36th Conference on Neural Information Processing Systems (NeurIPS 2022).

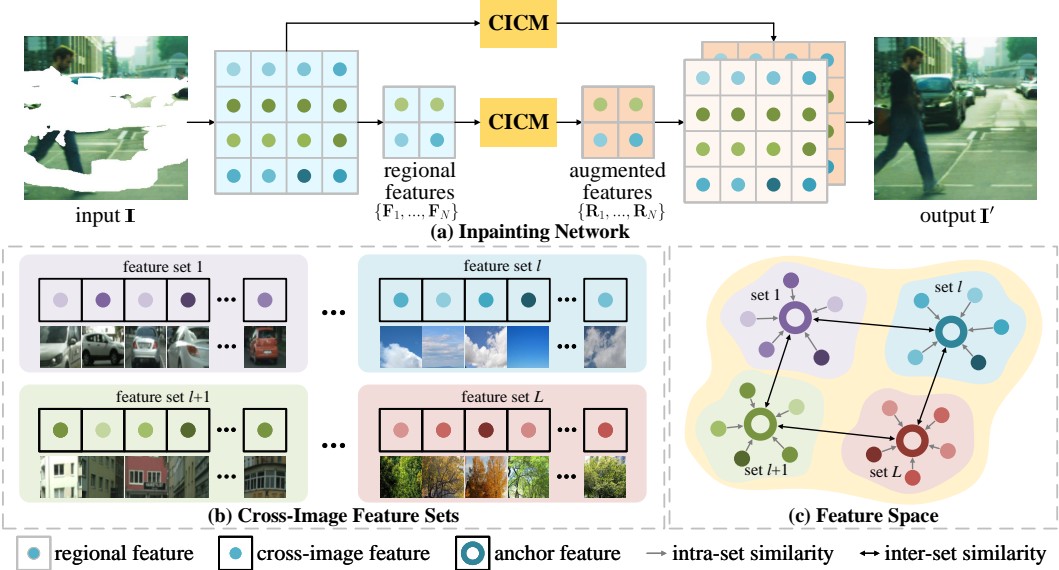

Figure 1: (a) CICM can be used to learn the cross-image features and augment the regional features at different layers of the inpainting network. (b) CICM consists of multiple cross-image feature sets, where each feature set contains several cross-image features. (c) In the feature space, we maximize/minimize the intra-/inter-set similarities between the cross-image features. It allows the cross-image features in different/identical set(s) to capture diverse/consistent context.

encourage the cross-image features to contain the visual information of the similar image regions. This facilitates *Context Augmentation* for finding the consistent cross-image features from CICM. Finally, we use the cross-image features to recover the corrupted regions.

We intensively evaluate CICM on the datasets (i.e., Places2 [9] and CelebA [10]) for image inpainting, where we achieve the state-of-the-art performances. Furthermore, we justify the generalization of CICM, which can be applied to learn the cross-image context from not only the low-level appearances but also the high-level semantic information of the image regions. With the help of CICM, we yield better results than the recent inpainting methods, on the datasets (i.e., Cityscapes [11] and Outdoor Scenes [12]) that provide the semantic information of object categories.

## 2  Related Work

### 2.1  Visual Context for Image Inpainting

Many inpainting methods have been proposed in recent years [13, 14, 15, 16, 17, 18, 19, 20, 21, 22]. Pathak et al. [23] propose the deep adversarial learning that harnesses visual context for image inpainting. Liu et al. [24] and Yu et al. [25] propose partial convolution and gated convolution to achieve visual context for recovering the corrupted image regions with irregular shapes. Liu et al. [26] focus on repairing the texture and structure of the corrupted images. Zeng et al. [27] fill the corrupted regions with the help of cross-layer attention. Li et al. [28] propose the recurrent feature reasoning to improve the inpainting results. In addition, semantic information has also been used to guide the recovery of the corrupted regions. Song et al. [29] use semantic segmentation maps for calibrating image inpainting. Liao et al. [30] employ the segmentation results at each layer of the decoder for constraining the inpainting results.

Most of the above methods rely on the single-image context. In contrast, we propose the inpainting method based on the cross-image context, which is more powerful in terms of capturing the underlying relationship between lost image contents and recovering the corrupted regions.

### 2.2  Memorized Context for Visual Understanding

Recent studies demonstrate the importance of the memorized context for the visual understanding tasks [31, 32, 33, 34, 35, 36, 37, 38]. He et al. [39] propose the momentum contrast to establish a large and consistent memory for unsupervised pre-training. Jeong et al. [40] propose the unsupervised

image-to-image translation framework that propagates the instance style information across images. Wang et al. [41] explore a large visual data space for memorizing the semantic object information. Xu et al. [42] design the memory that stores the complete regions for guiding texture generation. Feng et al. [43] model the semantic prior shared across images to reasonably recover the corrupted regions.

The memories of the above methods store the single-image context or the cross-image semantic information. They miss the appearance details shared across various images. In this paper, we construct the memory that contains the cross-image context learned from different images, allowing richer context to improve the inpainting results.

## 3 Method Overview

We illustrate the inpainting network with CICM in Figure 1. Given the input image $\mathbf{I} \in \mathbb{R}^{H \times W \times 3}$, the backbone network extracts the convolutional feature maps at different layers. At a layer, the feature map $\mathbf{F} \in \mathbb{R}^{H \times W \times C}$ is divided into $N$ regions, where we achieve the regional features $\{\mathbf{F}_n \in \mathbb{R}^C \mid n = 1, ..., N\}$. $H \times W$ and $C$ represent the spatial resolution and the feature channels.

CICM has the feature sets $\{\mathbb{S}_l \mid l = 1, ..., L\}$. $\mathbb{S}_l = \{\mathbf{C}_{l,k} \in \mathbb{R}^C \mid k = 1, ..., K\}$ is the $l^{th}$ set that contains $K$ cross-image features. $\mathbf{C}_{l,k}$ is the $k^{th}$ cross-image features. From $\{\mathbf{F}_n \in \mathbb{R}^C \mid n = 1, ..., N\}$, we choose the regional features of the complete regions and input them into the *Context Generalization*, which constructs the cross-image feature sets. The cross-image context provided by CICM is used by the *Context Augmentation* to recover the corrupted regions.

**Context Generalization** As illustrated in Figure 2(a), there is a set of anchor features $\{\mathbf{A}_l \in \mathbb{R}^C \mid l = 1, ..., L\}$ in CICM. $\mathbf{A}_l$ is associated with the feature set $\mathbb{S}_l$. Given the regional feature $\mathbf{F}_n$ of the complete region, we measure its similarity with each anchor feature. Given the anchor feature with the highest similarity, we choose the associated feature set, where we inject the new information of the regional feature $\mathbf{F}_n$ into the cross-image features. We also update the anchor feature to represent the entire feature set. During training, we maximize the intra-set similarity between the cross-image features in the same set. The inter-set similarity between the anchor features is minimized.

**Context Augmentation** CICM provides the cross-image features for the context augmentation, as illustrated in Figure 2(b). For the regional feature $\mathbf{F}_m$ of the corrupted region, we measure its similarity with every anchor feature. Again, we choose the anchor feature with the highest similarity, and use the cross-image features inside the associated feature set to augment the regional feature $\mathbf{F}_m$. This is done by injecting the context information of the cross-image features into $\mathbf{F}_m$. The augmented feature is used to predict the lost content of the region.

## 4 Cross-Image Context Memory

Below, we elaborate on the architecture of CICM. We use the *Context Generalization* to learn the cross-image features, which are stored in different feature sets of CICM. The *Context Augmentation* finds the cross-image features from CICM, offering the useful cross-image context for image inpainting.

### 4.1 Context Generalization

As illustrated in Figure 2(a), the context generalization extracts the regional features from the complete regions of different images. It injects these regional features into CICM, where the regional information is generalized as the cross-image features.

In Figure 2(c), we provide more details of the context generalization. We denote $\mathbf{F}_n$ as the regional feature of the $n^{th}$ region, which is complete in the input image $\mathbf{I}$. We follow the dot-product fashion to measure the similarity between the regional feature $\mathbf{F}_n$ and each anchor feature in the set $\{\mathbf{A}_l \in \mathbb{R}^C \mid l = 1, ..., L\}$, and choose the most relevant feature set $\mathbb{S}_l$ as:

$$l = \underset{o \in \{1, ..., L\}}{\arg\max} \mathbf{F}_n \cdot \mathbf{A}_o. \tag{1}$$

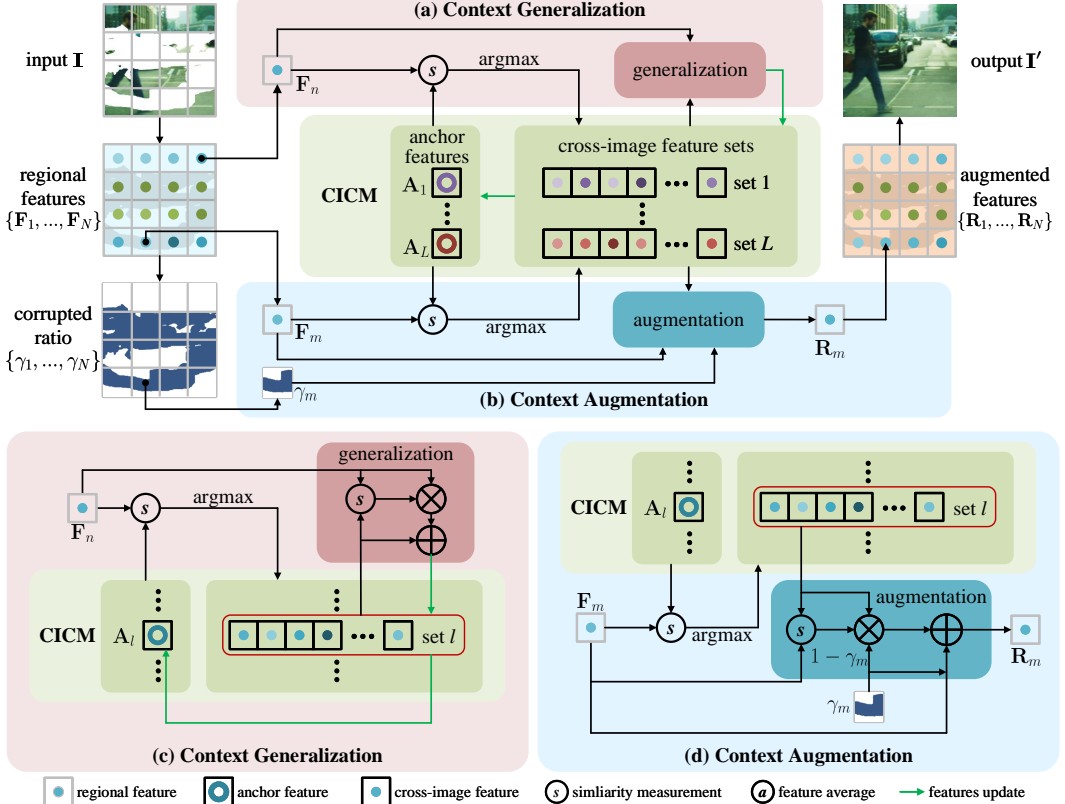

Figure 2: (a) Given a regional feature of the complete region, the context generation measures the similarity with every anchor feature, and selects the most similar anchor feature and the associated feature set. As illustrated in (c), the selected anchor feature and the cross-image features are updated by the regional information. (b) Given a regional feature of the corrupted region, the context augmentation also measures the similarity with every anchor feature. It select the feature set, where the cross-image features augment the regional feature, as illustrated in (d).

We inject the regional feature $\mathbf{F}_n$ into the set $\mathbb{S}_l$, where the cross-image feature $\mathbf{C}_{l,k}$ is updated as:

$$\mathbf{C}_{l,k} \leftarrow \mathbf{C}_{l,k} + \lambda_{n,l,k} \cdot \mathbf{F}_n,$$

$$s.t., \quad \lambda_{n,l,k} = \frac{\exp(\mathbf{F}_n \cdot \mathbf{C}_{l,k})}{\sum_{o=1}^{K} \exp(\mathbf{F}_n \cdot \mathbf{C}_{l,o})}, \quad \mathbf{C}_{l,k}, \mathbf{C}_{l,o} \in \mathbb{S}_l, \tag{2}$$

where $\leftarrow$ means the update by overwriting. We use the dot product to measure the similarity $\lambda_{n,l,k} \in [0,1]$ between the regional feature $\mathbf{F}_n$ and every cross-image feature in $\mathbb{S}_l$. The update of cross-image features take place during the network training. Note that the network training allows the regional features of different images to be injected into the cross-image features, whose generalization power is strengthened.

To improve the computational efficiency, we conduct the batch-wise update on the anchor features. After injecting the regional features of a mini-batch into the cross-image features in CICM, we update the anchor feature $\mathbf{A}_l$ as:

$$\mathbf{A}_l \leftarrow \beta \cdot \mathbf{A}_l + \frac{1-\beta}{K} \cdot \sum_{k=1}^{K} \mathbf{C}_{l,k}, \tag{3}$$

where we use the momentum factor $\beta = 0.5$ to control the update of the anchor feature.

## 4.2 Context Augmentation

As illustrated in Figure 2(b), the context augmentation takes input as the regional features of the corrupted regions in the image. The regional features are used to measure the similarities with the

anchor features. Based on the similarities for each regional feature, we find the most relevant anchor feature and the associate set of the cross-image features. We inject the generalized context of the cross-image features into the regional features of the corrupted regions, achieving the augmented features for recovering the lost contents.

The details of the context augmentation is illustrated in Figure 2(d). Given the regional feature $\mathbf{F}_m$ of the $m^{th}$ region, we measure its similarities with the anchor features, choosing the most similar anchor feature $\mathbf{A}_l$ and the associated feature set $\mathbb{S}_l$, as:

$$\mathbf{R}_m = \gamma_m \cdot \mathbf{F}_m + (1 - \gamma_m) \cdot \sum_{k=1}^{K} \lambda_{m,l,k} \cdot \mathbf{C}_{l,k},$$

$$s.t., \quad \lambda_{m,l,k} = \frac{\exp(\mathbf{F}_m \cdot \mathbf{C}_{l,k})}{\sum_{o=1}^{K} \exp(\mathbf{F}_m \cdot \mathbf{C}_{l,o})}, \quad l = \underset{o \in \{1,...,L\}}{\arg\max} \, \mathbf{F}_m \cdot \mathbf{A}_o, \quad \mathbf{C}_{l,k}, \mathbf{C}_{l,o} \in \mathbb{S}_l, \quad (4)$$

where we use all of the cross-image features in the set $\mathbb{S}_l$ to yield the augmented feature $\mathbf{R}_m \in \mathbb{R}^C$. This is done by weighting and adding the cross-image features $\{\mathbf{C}_{l,k} \in \mathbb{R}^C \mid k = 1, ..., K\}$ to $\mathbf{F}_m$.

The corrupted/complete status of the image region is unavailable for the inpainting task. Intuitively, the corrupted region requires rich context for reasoning the missing content. On the other hand, the complete region needs less context to preserve its specific property. Thus, for the $m^{th}$ region in the image, we input the regional feature $\mathbf{F}_m$ into the convolutional layer to estimate the corrupted ratio $\gamma_m \in [0, 1]$. As formulated in Eq. (4), the corrupted ratio is used to weight the regional feature and the cross-image features for computing the augmented feature. A larger ratio indicates a more corrupted region, whose regional representation is significantly augmented by the context information.

Finally, the augmented feature $\mathbf{R}_m$ is input into the convolutional layer for predicting the pixel values of the $m^{th}$ region. Based on the set of augmented features $\{\mathbf{R}_m \mid m = 1, ..., N\}$, we produce the inpainting result $\mathbf{I}' \in \mathbb{R}^{H \times W \times 3}$.

## 4.3 Training Objective

During the network training, we resort to L2-norm and the adversarial loss [44] for measuring the inpainting error as:

$$\mathcal{L}_{inpaint} = ||\mathbf{I}' - \bar{\mathbf{I}}||^2, \quad \mathcal{L}_{adv} = -\mathbb{E}_{\bar{\mathbf{I}}} \left[ \log(1 - \mathcal{D}(\bar{\mathbf{I}}, \mathbf{I}')) \right] - \mathbb{E}_{\mathbf{I}'} \left[ \log(\mathcal{D}(\mathbf{I}', \bar{\mathbf{I}})) \right], \quad (5)$$

where

$$\mathcal{D}(x, y) = \text{sigmoid}(\mathcal{C}(x) - \mathbb{E}_y \left[ \mathcal{C}(y) \right]). \quad (6)$$

In Eq.(5), $\bar{\mathbf{I}} \in \mathbb{R}^{H \times W \times 3}$ is the ground-truth result. $\mathbb{E}_x$ is the expectation with respect to $x$. In Eq.(6), $\mathcal{C}$ is a discriminator network. We also use L2-norm for measuring the estimated corrupted ratio and the ground-truth ratio of the image region as:

$$\mathcal{L}_{ratio} = \sum_{n=1}^{N} ||\gamma_n - \bar{\gamma}_n||^2, \quad (7)$$

where $\gamma_n$ and $\bar{\gamma}_n$ are the estimated and ground-truth corrupted ratios for the $n^{th}$ region in the input image $\mathbf{I}$. $\mathcal{L}_{ratio}$ is minimized. To achieve $\bar{\gamma}_n$, we subtract the input image $\mathbf{I}$ from the ground-truth result $\bar{\mathbf{I}}$, where the corrupted regions lead to larger differences of the pixel values. Within the $n^{th}$ region, we accumulate and normalize the differences of the pixel values, finally yielding $\bar{\gamma}_n$.

As formulated in Eq. (8), we use the dot product between the anchor features to measure the inter-set similarity, which is minimized to encourage different sets of cross-image features to capture more diverse context. In the same set, we maximize the intra-set similarity $\mathcal{L}_{intra}$ between the anchor feature and each cross-image feature, for encouraging the same set of cross-image features to capture more consistent context.

$$\mathcal{L}_{inter} = \sum_{l,o \in 1,...,L} \mathbf{A}_l \cdot \mathbf{A}_o, \quad \mathcal{L}_{intra} = \sum_{l \in 1,...,L} \sum_{k \in 1,...,K} \mathbf{A}_l \cdot \mathbf{C}_{l,k}. \quad (8)$$

With Eqs. (5)–(8), we formulate the overall training objective as:

$$\mathcal{L} = \lambda_1 \mathcal{L}_{inpaint} + \lambda_2 \mathcal{L}_{adv} + \lambda_3 \mathcal{L}_{ratio} + \lambda_4 \mathcal{L}_{inter} - \lambda_5 \mathcal{L}_{intra}, \quad (9)$$

where $\lambda_1 = 1.0$, $\lambda_2 = 0.1$, $\lambda_3 = 1.0$, $\lambda_4 = 20$ and $\lambda_5 = 0.5$. We minimize the overall objective $\mathcal{L}$ to optimize the network parameters.

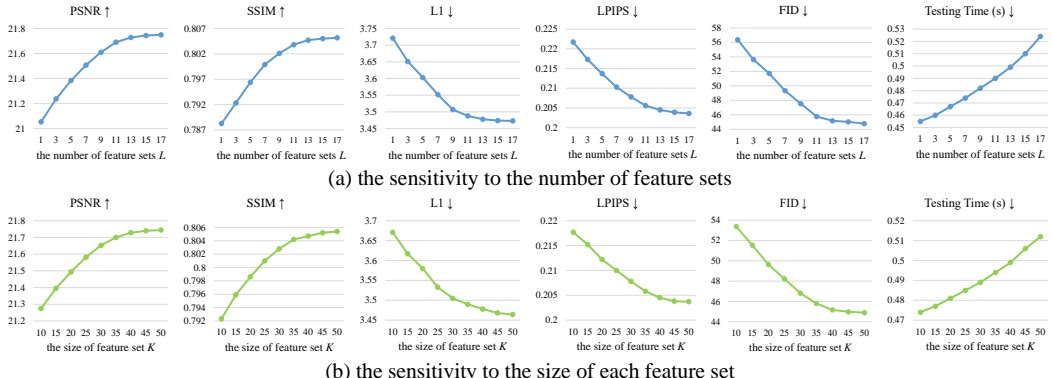

Figure 3: The results of sensitivity to memory capacity on the test set of Places2 with the mask of 20-40% area ratio. (a) We fix the size of each feature set as 40, and compare the inpainting performance by changing the number of feature sets $L$. (b) We fixed the number of feature sets as 13, and compare the inpainting performance by changing the size of each feature set.

## 5    Experiments

We evaluate our approach on the Places2 [9] and CelebA [10] datasets. Places2 provides over 8M images taken from over 365 scenes for training, along with 30K images for testing. It is used for the internal study of our approach. We also compare our approach with state-of-the-art methods on the CelebA dataset, which provides about 163K and 20K images, respectively, for training and testing.

We use the Cityscapes [11] and Outdoor Scenes [12] to extensively justify the effectiveness of our method. These datasets provide the semantic object categories for assisting inpainting task. Cityscapes contains 5,000 street-view images, where 2,975 and 1,525 images are used for training and testing. Outdoor Scenes contains 9,900 training and 300 testing images.

To compare the inpainting performances of different methods, the corrupted images are generated by the irregular masks [24, 25, 45]. We divided irregular masks into three groups, where 0-20%, 20-40%, and 40-60% of the image areas are corrupted. We employ the peak signal-to-noise ratio (PSNR), structural similarity index (SSIM), L1, perceptual similarity (LPIPS [46]), and fréchet inception distance (FID [47]) as the evaluation metrics. Note that PSNR, SSIM, L1, and FID measures the quality and realism of the restored image, while LPIPS measures the perceptual consistence between the restored image and the ground-truth image. Due to the limited space, we provide more details of the experimental setup in the supplementary material.

### 5.1    Internal Study

**Sensitivity to Memory Capacity**   The memory capacity of CICM is determined by the number of feature sets and the size of each set, which are set to 13 and 40 by default. Below, we change the memory capacity and examine the effect on the inpainting performance.

In Figure 3(a), we fix the size of each feature set as 40, and compare the inpainting performance by changing the number of feature sets $L$. We choose the number from the set $\{1, 3, 5, 7, 9, 11, 13, 15, 17\}$. With too less feature sets (e.g., $L = 1, 3, 5$), the diversity of cross-image context is unsatisfactorily captured, thus degrading the performance. On the other hand, too many feature sets (e.g., $L = 15, 17$) are redundant. They contain many cross-image features, which are computed based on similar image regions. These cross-image features little improve the performance but require extra testing time for each image.

Next, we change the size of each feature set (see Figure 3(b)). Here, the number of feature sets is fixed as 13. The size is chosen from the set $\{10, 15, 20, 25, 30, 35, 40, 45, 50\}$. In the large feature sets, a portion of the cross-image features loss the opportunity of update, thus containing many meaningless zeros. Compared to the well-learned features, the zero features unnecessarily occupy the memory space but have less impact on the inpainting performance.

**Variants of Context Generalization**   In Table 1 ("Context Generalization"), we evaluate the effectiveness of context generalization by comparing it with other alternatives.

We examine the impact of the intra- and inter-set similarities, which guide the learning of the cross-image features, on the context generalization. We remove either the intra- or inter-set similarity (see "w/o intra" and "w/o inter"), thus disallowing the cross-image features in different or identical sets to comprehensively capture the diverse or similar context. In this manner, we achieve lower performances than the full model trained with all of the similarities (see "intra & inter").

We use different ways of updating the cross-image features for context generalization. Given a regional feature, we find the most relevant feature set, where all of the cross-image features are updated (see "100% update"). We experiment with reducing the number of cross-image features to be updated. This is done by ranking the similarities between the regional feature and cross-image features, and selecting the top-50% (or even top-1) cross-image features for update. Fewer cross-image features for update reduce the testing time. But the cross-image features without updated likely contain outdated context information, letting themselves be inconsistent with the updated features in the same set. They largely degrade the performances (see "top-1 update" and "top-50% update").

| | Context Generalization | | | | | | | | | | | | | | |
|---|---|---|---|---|---|---|---|---|---|---|---|---|---|---|---|
| Methods | PSNR ↑ | | | SSIM ↑ | | | L1 ↓ | | | LPIPS ↓ | | | FID ↓ | | |
| | 0-20% | 20-40% | 40-60% | 0-20% | 20-40% | 40-60% | 0-20% | 20-40% | 40-60% | 0-20% | 20-40% | 40-60% | 0-20% | 20-40% | 40-60% |
| w/o intra | 28.032 | 20.711 | 16.647 | 0.9014 | 0.7566 | 0.5790 | 1.375 | 5.065 | 8.280 | 0.1277 | 0.2401 | 0.4369 | 22.75 | 53.75 | 129.3 |
| w/o inter | 27.954 | 20.384 | 16.338 | 0.8979 | 0.7521 | 0.5761 | 1.422 | 5.347 | 8.744 | 0.1243 | 0.2419 | 0.4423 | 24.36 | 55.73 | 132.7 |
| **intra & inter** | **29.214** | **21.728** | **19.211** | **0.9205** | **0.8047** | **0.6258** | **1.079** | **3.478** | **6.375** | **0.0829** | **0.2045** | **0.3284** | **17.21** | **45.17** | **78.49** |
| top-1 update | 28.021 | 20.309 | 16.632 | 0.9047 | 0.7692 | 0.5893 | 1.316 | 4.366 | 7.895 | 0.1186 | 0.2386 | 0.3918 | 21.86 | 59.73 | 126.4 |
| top-50% update | 28.413 | 20.923 | 17.118 | 0.9073 | 0.7731 | 0.5924 | 1.238 | 3.826 | 7.094 | 0.1017 | 0.2227 | 0.3642 | 20.57 | 56.39 | 106.4 |
| **100% update** | **29.214** | **21.728** | **19.211** | **0.9205** | **0.8047** | **0.6258** | **1.079** | **3.478** | **6.375** | **0.0829** | **0.2045** | **0.3284** | **17.21** | **45.17** | **78.49** |
| | Context Augmentation | | | | | | | | | | | | | | |
| Methods | PSNR ↑ | | | SSIM ↑ | | | L1 ↓ | | | LPIPS ↓ | | | FID ↓ | | |
| | 0-20% | 20-40% | 40-60% | 0-20% | 20-40% | 40-60% | 0-20% | 20-40% | 40-60% | 0-20% | 20-40% | 40-60% | 0-20% | 20-40% | 40-60% |
| w/o ratio | 28.551 | 21.217 | 18.879 | 0.9112 | 0.7983 | 0.6219 | 1.211 | 3.502 | 6.423 | 0.0891 | 0.2102 | 0.3310 | 19.02 | 55.21 | 85.48 |
| w/ ratio | 29.214 | 21.728 | 19.211 | 0.9205 | 0.8047 | 0.6258 | 1.079 | 3.478 | 6.375 | 0.0829 | 0.2057 | 0.3284 | 17.21 | 45.17 | 78.49 |
| **w/ GT** | **29.241** | **21.801** | **19.262** | **0.9227** | **0.8074** | **0.6281** | **1.074** | **3.462** | **6.356** | **0.0821** | **0.2040** | **0.3269** | **16.35** | **40.22** | **73.31** |
| top-1 aug | 28.317 | 19.878 | 17.866 | 0.9092 | 0.7882 | 0.6031 | 1.164 | 3.624 | 7.173 | 0.0921 | 0.2179 | 0.3531 | 20.81 | 72.78 | 95.64 |
| top-50% aug | 28.800 | 20.828 | 18.295 | 0.9152 | 0.7962 | 0.6115 | 1.122 | 3.576 | 6.724 | 0.0875 | 0.2115 | 0.3376 | 18.83 | 63.32 | 87.35 |
| **100% aug** | **29.214** | **21.728** | **19.211** | **0.9205** | **0.8047** | **0.6258** | **1.079** | **3.478** | **6.375** | **0.0829** | **0.2045** | **0.3284** | **17.21** | **45.17** | **78.49** |

Table 1: The results of various generalization and augmentation ways on the test set of Places2.

**Variants of Context Augmentation**   Given the cross-image features, we experiment with using various ways of context augmentation, and report the results in Table 1 ("Context Augmentation").

For the context augmentation, we estimate the corrupted ratios for controlling the context information, which is propagated from the cross-image features to the regional features. We experiment with setting the corrupted ratios to 1 unanimously. It means that the context information is fully propagated to every regional feature. The regional features of the complete regions are heavily affected by the cross-image features, yielding lower performances (see "w/o ratio"). This is because the diverse cross-image information is largely involved into the regional features, which are supposed to keep their original information for preserving the appearances of the complete regions in the result. We also compare the inpainting performances, which are achieved based on the estimated and ground-truth corrupted ratios (see "w/ ratio" and "w/ GT"). Though the ground-truth corrupted ratios are more accurate than the estimated ones, the gap between the inpainting performances is reasonable. It demonstrates the robustness of estimating the corrupted ratios.

Moreover, we study the impact of changing the number of the cross-image features, which are used by the context augmentation of the regional features. Note that our full model resorts to all of the cross-image features in the relevant set for feature augmentation (see "100% aug"). Based on the similarities between the cross-image features and the regional features, we select the top-1 and top-50% of the cross-image features, respectively, for augmenting the regional features. Apparently, it disallows the regional features of the corrupted regions to benefit from richer cross-image context, consequently leading to worse inpainting results.

| Single-Image Context | | | | | | | | | | | | | | | |
|---|---|---|---|---|---|---|---|---|---|---|---|---|---|---|---|
| Methods | PSNR ↑ | | | SSIM ↑ | | | L1 ↓ | | | LPIPS ↓ | | | FID ↓ | | |
| | 0-20% | 20-40% | 40-60% | 0-20% | 20-40% | 40-60% | 0-20% | 20-40% | 40-60% | 0-20% | 20-40% | 40-60% | 0-20% | 20-40% | 40-60% |
| k-means | 28.311 | 20.946 | 17.021 | 0.9092 | 0.7734 | 0.5882 | 1.217 | 4.275 | 7.302 | 0.1049 | 0.2369 | 0.3992 | 20.21 | 61.37 | 110.4 |
| RUC | 28.386 | 20.997 | 17.105 | 0.9104 | 0.7763 | 0.5921 | 1.194 | 4.113 | 7.189 | 0.1025 | 0.2346 | 0.3927 | 19.46 | 67.71 | 106.6 |
| anchor only | 28.417 | 21.015 | 17.235 | 0.9120 | 0.7828 | 0.5977 | 1.164 | 3.872 | 7.071 | 0.0998 | 0.2320 | 0.3875 | 18.67 | 56.52 | 102.5 |

| Cross-Image Context | | | | | | | | | | | | | | | |
|---|---|---|---|---|---|---|---|---|---|---|---|---|---|---|---|
| Methods | PSNR ↑ | | | SSIM ↑ | | | L1 ↓ | | | LPIPS ↓ | | | FID ↓ | | |
| | 0-20% | 20-40% | 40-60% | 0-20% | 20-40% | 40-60% | 0-20% | 20-40% | 40-60% | 0-20% | 20-40% | 40-60% | 0-20% | 20-40% | 40-60% |
| merged sets | 28.833 | 21.054 | 17.574 | 0.9130 | 0.7883 | 0.6035 | 1.145 | 3.721 | 6.857 | 0.0882 | 0.2217 | 0.3638 | 19.23 | 53.44 | 96.47 |
| **CICM** | **29.214** | **21.728** | **19.211** | **0.9205** | **0.8047** | **0.6258** | **1.079** | **3.478** | **6.375** | **0.0829** | **0.2045** | **0.3284** | **17.21** | **45.17** | **78.49** |

Table 2: The results of various ways of using context information on the test set of Places2.

**Different Ways of Using Image Context**  In Table 2, we compare CICM with different ways of using image context for image inpainting. Here, we divide the methods into two groups, which respectively use the single- and cross-image context for image inpainting.

In Table 2 ("Single-Image Context"), we report the results of using k-means and the deep-learning-based RUC [48], which are the clustering methods for harnessing the single-image context in our scenario. For a single image, we use the clustering methods to divide the regional features of the complete regions into several clusters. Based on the similarities with the cluster centers, we find the most relevant regional features for recovering the corrupted regions. Another variant of using the single-image context is to remove the feature sets of CICM and keep the anchor features only. The anchor features play in place of the cluster centers, which are produced by the clustering methods. Compared to the cross-image context, the single-image context (see "k-means", "RUC" and "anchor only") leads to lower performances.

In Table 2 ("Cross-Image Context"), we experiment with merging the feature sets constructed by the context generalization. This is done by removing the anchor features, only relying on the similarities between the cross-image features and the regional features for feature augmentation. This method use the cross-image features, which contain discrepant visual context, to recover each corrupted region. Even though too discrepant visual context can be filtered by the lower feature similarity, many cross-image features with diverse information still distract the inpainting of a corrupted region. It leads to worse results than CICM, which employs the anchor features to select the relevant context.

| Places2 Dataset | | | | | | | | | | | | | | | |
|---|---|---|---|---|---|---|---|---|---|---|---|---|---|---|---|
| Methods | PSNR ↑ | | | SSIM ↑ | | | L1 ↓ | | | LPIPS ↓ | | | FID ↓ | | |
| | 0-20% | 20-40% | 40-60% | 0-20% | 20-40% | 40-60% | 0-20% | 20-40% | 40-60% | 0-20% | 20-40% | 40-60% | 0-20% | 20-40% | 40-60% |
| UNet | 28.637 | 20.944 | 17.022 | 0.9141 | 0.7885 | 0.5746 | 1.137 | 3.606 | 7.269 | 0.0850 | 0.2162 | 0.3838 | 18.37 | 58.22 | 112.7 |
| **UNet-CICM** | **29.214** | **21.728** | **18.811** | **0.9205** | **0.8047** | **0.6258** | **1.079** | **3.478** | **6.375** | **0.0829** | **0.2045** | **0.3284** | **17.21** | **45.17** | **78.49** |
| RFR [28] | 28.891 | 21.278 | 17.648 | 0.9167 | 0.7893 | 0.5953 | 1.128 | 3.532 | 6.916 | 0.0873 | 0.2267 | 0.3723 | 17.83 | 51.29 | 95.72 |
| **RFR-CICM** | **29.411** | **22.146** | **19.313** | **0.9210** | **0.8134** | **0.6311** | **1.065** | **3.337** | **6.211** | **0.0834** | **0.2088** | **0.3174** | **16.69** | **40.23** | **64.17** |
| JPG [49] | 30.023 | 22.561 | 18.045 | 0.9362 | 0.8267 | 0.6762 | 0.902 | 2.671 | 5.725 | 0.0883 | 0.2417 | 0.3521 | 16.78 | 39.21 | 78.77 |
| **JPG-CICM** | **30.457** | **23.716** | **20.016** | **0.9417** | **0.8325** | **0.7022** | **0.868** | **2.516** | **5.073** | **0.0835** | **0.2174** | **0.3093** | **16.02** | **34.88** | **58.19** |
| MISF [50] | 31.044 | 23.799 | 19.314 | 0.9443 | 0.8312 | 0.6736 | 0.741 | 2.520 | 5.311 | 0.0537 | 0.1721 | 0.2821 | 16.39 | 35.31 | 62.67 |
| **MISF-CICM** | **31.516** | **24.858** | **21.267** | **0.9491** | **0.8405** | **0.7027** | **0.712** | **2.317** | **4.872** | **0.0501** | **0.1498** | **0.2389** | **14.76** | **29.12** | **48.21** |

| CelebA Dataset | | | | | | | | | | | | | | | |
|---|---|---|---|---|---|---|---|---|---|---|---|---|---|---|---|
| Methods | PSNR ↑ | | | SSIM ↑ | | | L1 ↓ | | | LPIPS ↓ | | | FID ↓ | | |
| | 0-20% | 20-40% | 40-60% | 0-20% | 20-40% | 40-60% | 0-20% | 20-40% | 40-60% | 0-20% | 20-40% | 40-60% | 0-20% | 20-40% | 40-60% |
| UNet | 33.133 | 24.573 | 19.522 | 0.9577 | 0.8621 | 0.7234 | 0.533 | 1.882 | 4.623 | 0.0432 | 0.1282 | 0.2476 | 10.74 | 40.83 | 75.39 |
| **UNet-CICM** | **33.388** | **25.384** | **21.673** | **0.9610** | **0.8788** | **0.7689** | **0.518** | **1.820** | **4.127** | **0.0419** | **0.1214** | **0.2238** | **8.493** | **35.47** | **63.22** |
| RFR [28] | 33.327 | 25.224 | 20.133 | 0.9571 | 0.8722 | 0.7323 | 0.538 | 1.872 | 4.638 | 0.0437 | 0.1257 | 0.2421 | 9.362 | 33.28 | 67.31 |
| **RFR-CICM** | **33.636** | **26.056** | **21.517** | **0.9601** | **0.8793** | **0.7654** | **0.515** | **1.784** | **4.164** | **0.0408** | **0.1166** | **0.2287** | **7.134** | **31.78** | **56.98** |
| JPG [49] | 33.925 | 26.338 | 20.548 | 0.9573 | 0.8826 | 0.7428 | 0.527 | 1.692 | 4.411 | 0.0427 | 0.1307 | 0.2559 | 8.273 | 32.02 | 61.32 |
| **JPG-CICM** | **34.262** | **27.027** | **22.393** | **0.9619** | **0.8902** | **0.7681** | **0.504** | **1.646** | **3.817** | **0.0401** | **0.1186** | **0.2265** | **6.374** | **29.26** | **53.87** |
| MISF [50] | 34.302 | 26.387 | 21.289 | 0.9629 | 0.8903 | 0.7585 | 0.501 | 1.572 | 3.922 | 0.0336 | 0.0981 | 0.2137 | 6.836 | 30.11 | 55.75 |
| **MISF-CICM** | **34.695** | **27.854** | **23.338** | **0.9683** | **0.9012** | **0.7782** | **0.489** | **1.502** | **3.311** | **0.0317** | **0.0925** | **0.1921** | **5.023** | **27.99** | **47.12** |

Table 3: The results of combining CICM with different inpainting networks (i.e., RFR [28], JPG [49], and MISF [50]) on the test sets of Places2 and CelebA.

## 5.2 External Evaluation

**Combination with Different Inpainting Networks**  To justify the generalization power of CICM, we equip CICM with different inpainting networks and report the change of performances in Table 3. Apart from the baseline model, we choose RFR [28], JPG [49], and MISF [50] for evaluation. For a fair comparison, these methods use the lowest-resolution convolution feature maps to extract the regional features, which are used for learning the cross-image features in CICM. With CICM, different methods consistently improve the inpainting results on the test sets of Places2 and CelebA.

**Comparison with State-of-the-Art Methods**  In Table 4, we compare our method with state-of-the-art methods [28, 49, 50, 42, 43] on the test sets of Places2 and CelebA. We use a UNet with 15 convolutional layers as the backbone. Among the compared methods, TMAD [42] and SRM [43] also construct the memory banks, which store the useful context for recovering the corrupted regions. In contrast to CICM, the memory bank of TMAD contains the low-level features (e.g., colors and textures) of the complete regions in a single image; the memory bank of SRM provides the cross-image distributions of semantic categories, but missing the appearance information of the image regions. In comparison, CICM achieves better performances than other methods. We visualized the results of different methods in Figure 4(a).

| Places2 Dataset | | | | | | | | | | | | | | |
|---|---|---|---|---|---|---|---|---|---|---|---|---|---|---|
| Methods | PSNR ↑ | | | SSIM ↑ | | | L1 ↓ | | | LPIPS ↓ | | | FID ↓ | | |
| | 0-20% | 20-40% | 40-60% | 0-20% | 20-40% | 40-60% | 0-20% | 20-40% | 40-60% | 0-20% | 20-40% | 40-60% | 0-20% | 20-40% | 40-60% |
| RFR [28] | 29.281 | 22.589 | 18.581 | 0.9283 | 0.7868 | 0.6137 | 1.009 | 3.218 | 6.719 | 0.0825 | 0.2161 | 0.3571 | 20.47 | 47.21 | 82.11 |
| JPG [49] | 30.673 | 23.937 | 19.884 | 0.9452 | 0.8348 | 0.6915 | 0.830 | 2.581 | 5.294 | 0.0817 | 0.2145 | 0.3235 | 19.48 | 42.29 | 74.41 |
| MISF [50] | 31.335 | 24.239 | 20.044 | 0.9506 | 0.8435 | 0.6931 | 0.726 | 2.340 | 4.965 | 0.0432 | 0.1498 | 0.2499 | 17.46 | 37.28 | 69.47 |
| TMAD [42] | 30.273 | 22.185 | 19.172 | 0.9352 | 0.8169 | 0.6577 | 0.813 | 2.561 | 5.399 | 0.0612 | 0.1438 | 0.2556 | 21.55 | 49.33 | 93.41 |
| SRM [43] | 31.434 | 24.685 | 21.021 | 0.9523 | 0.8462 | 0.7021 | 0.732 | 2.217 | 4.937 | 0.0438 | 0.1253 | 0.2218 | 15.78 | 32.13 | 66.34 |
| **CICM** | **32.237** | **26.098** | **22.549** | **0.9573** | **0.8665** | **0.7323** | **0.689** | **2.168** | **4.526** | **0.0406** | **0.1216** | **0.2064** | **13.47** | **27.38** | **54.47** |

| CelebA Dataset | | | | | | | | | | | | | | |
|---|---|---|---|---|---|---|---|---|---|---|---|---|---|---|---|
| Methods | PSNR ↑ | | | SSIM ↑ | | | L1 ↓ | | | LPIPS ↓ | | | FID ↓ | | |
| | 0-20% | 20-40% | 40-60% | 0-20% | 20-40% | 40-60% | 0-20% | 20-40% | 40-60% | 0-20% | 20-40% | 40-60% | 0-20% | 20-40% | 40-60% |
| RFR [28] | 33.728 | 25.711 | 20.721 | 0.9633 | 0.8762 | 0.7431 | 0.515 | 1.799 | 4.319 | 0.0395 | 0.1185 | 0.2301 | 8.236 | 30.14 | 68.28 |
| JPG [49] | 34.492 | 26.623 | 21.365 | 0.9691 | 0.8915 | 0.7715 | 0.473 | 1.639 | 4.025 | 0.0429 | 0.1295 | 0.2472 | 6.438 | 25.98 | 62.88 |
| MISF [50] | 34.604 | 26.693 | 21.638 | 0.9702 | 0.8923 | 0.7721 | 0.471 | 1.603 | 3.802 | 0.0311 | 0.0938 | 0.1889 | 6.125 | 22.31 | 53.55 |
| TMAD [42] | 33.891 | 25.751 | 20.388 | 0.9532 | 0.8783 | 0.7521 | 0.544 | 1.691 | 4.349 | 0.0356 | 0.1143 | 0.2126 | 13.39 | 46.90 | 80.21 |
| SRM [43] | 34.263 | 26.815 | 21.723 | 0.9645 | 0.8954 | 0.7754 | 0.483 | 1.584 | 3.632 | 0.0318 | 0.0954 | 0.1846 | 4.426 | 20.54 | 48.24 |
| **CICM** | **34.895** | **27.925** | **23.638** | **0.9733** | **0.9087** | **0.7947** | **0.466** | **1.462** | **3.197** | **0.0294** | **0.0880** | **0.1746** | **4.115** | **18.31** | **43.51** |

Table 4: Comparison with state-of-the-art methods on the test sets of Places2 and CelebA.

**Extensive Evaluation on Semantic Inpainting**  We extensively evaluate CICM on the public datasets (i.e., Cityscapes and Outdoor Scenes), which provide the semantic object categories for assisting the image inpainting task. Here, CICM can be easily extended to learn the cross-image features from not only the RGB images but also the semantic segmentation results. On the test sets of Cityscapes and Outdoor Scenes, CICM outperforms state-of-the-art methods [25, 27, 29, 30, 51], thus demonstrating its power in terms of learning the cross-image context from various data sources. The visualized results of different methods are shown in Figure 4(b).

## 6  Conclusion

The latest progress in image inpainting benefits from the visual context learned by deep neural networks for representing the relationship between the image regions. It helps to propagate the information of the complete image regions to recover the corrupted regions. In this paper, we propose the *Cross-Image Context Memory* (CICM), where the visual context is learned across different images. Compared to the methods that utilize the single-image context for inpainting, CICM better addresses the problem of missing the intrinsic relationship between regions for learning the visual context, given the images with lost contents. CICM achieves state-of-the-art performances on the public datasets.

In the future, we plan to extend CICM to other tasks (e.g., 3D scene completion and occluded object detection), where the critical image contents are required to be recovered and understood.

| Cityscapes Dataset | | | | | | | | | | | | | | | |
|---|---|---|---|---|---|---|---|---|---|---|---|---|---|---|---|
| Methods | PSNR ↑ | | | SSIM ↑ | | | L1 ↓ | | | LPIPS ↓ | | | FID ↓ | | |
| | 0-20% | 20-40% | 40-60% | 0-20% | 20-40% | 40-60% | 0-20% | 20-40% | 40-60% | 0-20% | 20-40% | 40-60% | 0-20% | 20-40% | 40-60% |
| Gated [25] | 35.870 | 27.011 | 22.938 | 0.965 | 0.801 | 0.748 | 0.518 | 2.627 | 3.872 | 0.0345 | 0.0576 | 0.1573 | 5.298 | 23.28 | 52.29 |
| PEN [27] | 33.693 | 23.927 | 22.336 | 0.964 | 0.800 | 0.694 | 0.548 | 3.132 | 4.012 | 0.0317 | 0.0552 | 0.1662 | 8.314 | 48.67 | 66.70 |
| SPG [29] | 29.627 | 25.425 | 21.863 | 0.900 | 0.817 | 0.718 | 0.722 | 2.877 | 4.188 | 0.0412 | 0.0610 | 0.1627 | 17.14 | 27.09 | 36.42 |
| SWAP [30] | 32.973 | 26.112 | 22.984 | 0.965 | 0.885 | 0.782 | 0.602 | 2.435 | 3.762 | 0.0365 | 0.0547 | 0.1543 | 6.327 | 15.48 | 29.32 |
| SPL [51] | 35.543 | 27.639 | 23.530 | 0.969 | 0.892 | 0.773 | 0.476 | 2.192 | 3.353 | 0.0311 | 0.0486 | 0.1263 | 4.686 | 12.94 | 28.81 |
| **CICM** | **36.728** | **29.848** | **25.625** | **0.978** | **0.912** | **0.844** | **0.421** | **1.748** | **2.943** | **0.0281** | **0.0465** | **0.1170** | **3.437** | **8.246** | **12.16** |

| Outdoor Scenes Dataset | | | | | | | | | | | | | | | |
|---|---|---|---|---|---|---|---|---|---|---|---|---|---|---|---|
| Methods | PSNR ↑ | | | SSIM ↑ | | | L1 ↓ | | | LPIPS ↓ | | | FID ↓ | | |
| | 0-20% | 20-40% | 40-60% | 0-20% | 20-40% | 40-60% | 0-20% | 20-40% | 40-60% | 0-20% | 20-40% | 40-60% | 0-20% | 20-40% | 40-60% |
| Gated [25] | 30.826 | 24.262 | 19.294 | 0.955 | 0.874 | 0.680 | 0.910 | 2.018 | 2.684 | 0.0778 | 0.1578 | 0.2320 | 20.23 | 49.42 | 89.84 |
| PEN [27] | 29.072 | 21.515 | 19.237 | 0.949 | 0.796 | 0.630 | 1.036 | 2.438 | 2.711 | 0.0781 | 0.1662 | 0.2533 | 19.82 | 59.78 | 90.21 |
| SPG [29] | 24.156 | 21.692 | 18.282 | 0.801 | 0.685 | 0.533 | 1.417 | 2.638 | 2.764 | 0.0865 | 0.2218 | 0.2764 | 46.48 | 72.96 | 101.3 |
| SWAP [30] | 30.361 | 25.116 | 20.832 | 0.948 | 0.861 | 0.702 | 0.832 | 1.866 | 2.495 | 0.0572 | 0.1683 | 0.2217 | 13.29 | 40.01 | 63.86 |
| SPL [51] | 32.599 | 25.485 | 21.083 | 0.961 | 0.864 | 0.710 | 0.749 | 1.729 | 2.387 | 0.0465 | 0.1304 | 0.2042 | 11.24 | 30.07 | 53.28 |
| **CICM** | **33.271** | **26.467** | **22.116** | **0.969** | **0.886** | **0.732** | **0.674** | **1.411** | **2.011** | **0.0412** | **0.1141** | **0.1872** | **8.684** | **23.35** | **42.47** |

Table 5: Comparison with state-of-the-art methods on the test sets of Cityscapes and Outdoor Scenes.

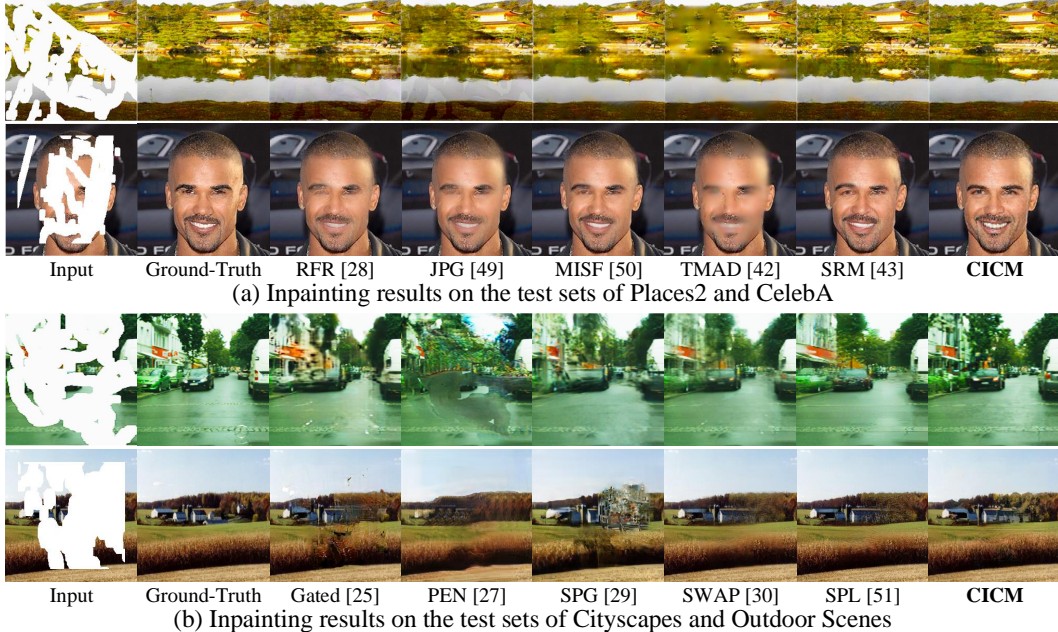

(a) Inpainting results on the test sets of Places2 and CelebA

(b) Inpainting results on the test sets of Cityscapes and Outdoor Scenes

Figure 4: We provide the inpainting results of different methods. The images are taken from on (a) Places2 (top) and CelebA (bottom) and (b) Cityscapes (top) and Outdoor Scenes (bottom).

## Negative Societal Impacts

Our approach can help to recover the image information, which may be broadly used in many scenarios (e.g., autonomous vehicle and video surveillance). One should be cautious of using the results, which may contain problematic information. This may give rise to the infringement of privacy or economic interest.

## Acknowledgments

We thank the anonymous reviewers for their constructive suggestions. This work is supported in parts by the National Key Research and Development Program of China (2020YFC1522701) and the National Natural Science Foundation of China (62072334).

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
