# Cross-Image Context for Single Image Inpainting
## – Supplementary Material –

**Tingliang Feng,   Wei Feng,   Weiqi Li,   Di Lin**[*]
College of Intelligence and Computing, Tianjin University
{fengtl, wicky}@tju.edu.cn, wfeng@ieee.org, Ande.lin1988@gmail.com

## 1   Implementation Details

We use the PyTorch toolkit to implement our inpainting network with CICM. The network is optimized by the Adam solver for 400,000 iterations. The initial learning rate is 0.0001, which is linearly decayed during the network training. Each mini-batch contains 8 images with the size of $256 \times 256$. We randomly crop and flip the training images to augment the data. The network is trained on 4 RTX 2080Ti GPUs.

In our implementation, we use a warm-up strategy to pre-train the backbone network for 50,000 iterations. The encoder of the pre-trained backbone is used to compute the regional features of different images. We conduct k-means clustering on the regional features, computing the cluster centers as the initial anchor features. The regional features, which are nearest to the initial anchor features, are selected as the initial cross-image features in different sets of CICM.

To augment the training data, we generate the corrupted regions in the training images, by randomly matching the irregular masks [1, 2, 3] and the RGB images. For a fair comparison during the network testing, the corrupted regions in the testing images are fixed for different methods.

## 2   Supplementary Experiments

In this section, we provide more details of the experiments. We divide this section into four parts. The first part is a supplementary description of experimental setups in the main paper. The second part is the convergence analysis of our network training. The third part is the analysis on distributions of cross-image features. The fourth part is to show more visual results on different datasets.

### 2.1   Supplementary Description of Experimental Setup

**Variants of Context Generalization**   In Table 1 ("Context Generalization"), which has been presented in the main paper, we use different ways of updating the cross-image features for context generalization. Given a regional feature, we find the most relevant feature set, where all of the cross-image features are updated (see "100% update"). We experiment with reducing the number of cross-image features to be updated. This is done by ranking the similarities between the regional feature and cross-image features, and selecting the top-50% (or even top-1) cross-image features for updating.

After calculating the similarities between the regional feature $\mathbf{F}_n$ and the cross-image features in the chosen feature set, we need to update the cross-image features for context generalization. Here, we use three different ways of updating, as illustrated in Figure 1. In Figure 1(a), we only update the cross-image feature that has the highest similarity with the regional feature $\mathbf{F}_n$. In Figure 1(b), we update the top-50% cross-image features according to the similarities. In Figure 1(c), we update all the cross-image features in the chosen feature set.

---

[*]Di Lin is the corresponding author of this paper.

36th Conference on Neural Information Processing Systems (NeurIPS 2022).

| | Context Generalization | | | | | | | | | | | | | | |
|---|---|---|---|---|---|---|---|---|---|---|---|---|---|---|---|
| Methods | PSNR ↑ | | | SSIM ↑ | | | L1 ↓ | | | LPIPS ↓ | | | FID ↓ | | |
| | 0-20% | 20-40% | 40-60% | 0-20% | 20-40% | 40-60% | 0-20% | 20-40% | 40-60% | 0-20% | 20-40% | 40-60% | 0-20% | 20-40% | 40-60% |
| w/o intra | 28.032 | 20.711 | 16.647 | 0.9014 | 0.7566 | 0.5790 | 1.375 | 5.065 | 8.280 | 0.1277 | 0.2401 | 0.4369 | 22.75 | 53.75 | 129.3 |
| w/o inter | 27.954 | 20.384 | 16.338 | 0.8979 | 0.7521 | 0.5761 | 1.422 | 5.347 | 8.744 | 0.1243 | 0.2419 | 0.4423 | 24.36 | 55.73 | 132.7 |
| **intra & inter** | **29.214** | **21.728** | **19.211** | **0.9205** | **0.8047** | **0.6258** | **1.079** | **3.478** | **6.375** | **0.0829** | **0.2045** | **0.3284** | **17.21** | **45.17** | **78.49** |
| top-1 update | 28.021 | 20.309 | 16.632 | 0.9047 | 0.7692 | 0.5893 | 1.316 | 4.366 | 7.895 | 0.1186 | 0.2386 | 0.3918 | 21.86 | 59.73 | 126.4 |
| top-50% update | 28.413 | 20.923 | 17.118 | 0.9073 | 0.7731 | 0.5924 | 1.238 | 3.826 | 7.094 | 0.1017 | 0.2227 | 0.3642 | 20.57 | 56.39 | 106.4 |
| **100% update** | **29.214** | **21.728** | **19.211** | **0.9205** | **0.8047** | **0.6258** | **1.079** | **3.478** | **6.375** | **0.0829** | **0.2045** | **0.3284** | **17.21** | **45.17** | **78.49** |

| | Context Augmentation | | | | | | | | | | | | | | |
|---|---|---|---|---|---|---|---|---|---|---|---|---|---|---|---|
| Methods | PSNR ↑ | | | SSIM ↑ | | | L1 ↓ | | | LPIPS ↓ | | | FID ↓ | | |
| | 0-20% | 20-40% | 40-60% | 0-20% | 20-40% | 40-60% | 0-20% | 20-40% | 40-60% | 0-20% | 20-40% | 40-60% | 0-20% | 20-40% | 40-60% |
| w/o ratio | 28.551 | 21.217 | 18.879 | 0.9112 | 0.7983 | 0.6219 | 1.211 | 3.502 | 6.423 | 0.0891 | 0.2102 | 0.3310 | 19.02 | 55.21 | 85.48 |
| w/ ratio | 29.214 | 21.728 | 19.211 | 0.9205 | 0.8047 | 0.6258 | 1.079 | 3.478 | 6.375 | 0.0829 | 0.2057 | 0.3284 | 17.21 | 45.17 | 78.49 |
| **w/ GT** | **29.241** | **21.801** | **19.262** | **0.9227** | **0.8074** | **0.6281** | **1.074** | **3.462** | **6.356** | **0.0821** | **0.2040** | **0.3269** | **16.35** | **40.22** | **73.31** |
| top-1 aug | 28.317 | 19.878 | 17.866 | 0.9092 | 0.7882 | 0.6031 | 1.164 | 3.624 | 7.173 | 0.0921 | 0.2179 | 0.3531 | 20.81 | 72.78 | 95.64 |
| top-50% aug | 28.800 | 20.828 | 18.295 | 0.9152 | 0.7962 | 0.6115 | 1.122 | 3.576 | 6.724 | 0.0875 | 0.2115 | 0.3376 | 18.83 | 63.32 | 87.35 |
| **100% aug** | **29.214** | **21.728** | **19.211** | **0.9205** | **0.8047** | **0.6258** | **1.079** | **3.478** | **6.375** | **0.0829** | **0.2045** | **0.3284** | **17.21** | **45.17** | **78.49** |

Table 1: The results of various generalization and augmentation ways on the test set of Places2.

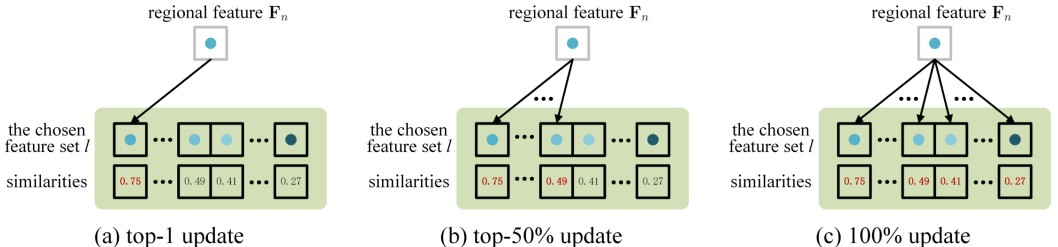

(a) top-1 update    (b) top-50% update    (c) 100% update

Figure 1: Three different ways of updating the cross-image features in the chosen feature set according to the similarities with the regional feature $\mathbf{F}_n$. (a) Updating the cross-image feature with the highest similarity. (b) Updating the top-50% cross-image features. (c) Updating all the cross-image features.

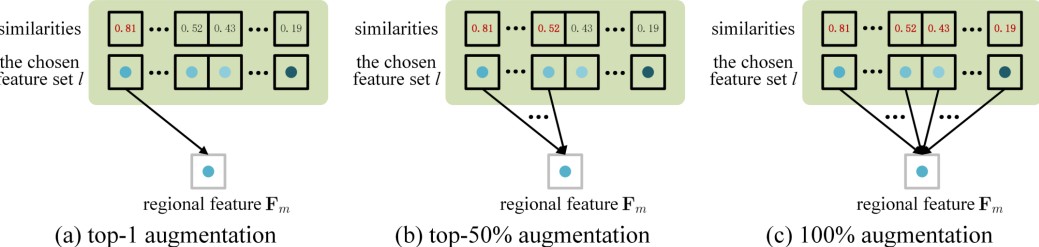

(a) top-1 augmentation    (b) top-50% augmentation    (c) 100% augmentation

Figure 2: Three different ways of the regional feature $\mathbf{F}_m$ augmentation according to the similarities with the cross-image features in the chosen feature set. (a) Augmentation by using the cross-image feature with the highest similarity. (b) Augmentation by using the top-50% cross-image features. (c) Augmentation by using all cross-image features.

**Variants of Context Augmentation**  In Table 1 ("Context Augmentation"), which has been presented in the main paper, we study the impact of changing the number of the cross-image features, which are used by the context augmentation of the regional features. Note that our full model resorts to all of the cross-image features in the relevant set for feature augmentation (see "100% aug"). Based on the similarities between the cross-image features and the regional features, we select the top-1 and top-50% of the cross-image features, respectively, for augmenting the regional features. We illustrate these variants of context augmentation in Figure 2.

In Figure 2(a), we only choose the cross-image feature with the highest similarity to augment the regional feature $\mathbf{F}_m$. In Figure 2(b), we use the top-50% cross-image features according to the similarities for augmentation. In Figure 2(c), the whole cross-image features in the chosen feature set are used for context augmentation.

**Different Ways of Using Image Context**  In Table 2 ("Single-Image Context"), which has been presented in the main paper, we report the results of using k-means and the deep-learning-based

| | PSNR ↑ | | | SSIM ↑ | | | L1 ↓ | | | LPIPS ↓ | | | FID ↓ | | |
|---|---|---|---|---|---|---|---|---|---|---|---|---|---|---|---|
| **Single-Image Context** | | | | | | | | | | | | | | | |
| Methods | 0-20% | 20-40% | 40-60% | 0-20% | 20-40% | 40-60% | 0-20% | 20-40% | 40-60% | 0-20% | 20-40% | 40-60% | 0-20% | 20-40% | 40-60% |
| k-means | 28.311 | 20.946 | 17.021 | 0.9092 | 0.7734 | 0.5882 | 1.217 | 4.275 | 7.302 | 0.1049 | 0.2369 | 0.3992 | 20.21 | 61.37 | 110.4 |
| RUC | 28.386 | 20.997 | 17.105 | 0.9104 | 0.7763 | 0.5921 | 1.194 | 4.113 | 7.189 | 0.1025 | 0.2346 | 0.3927 | 19.46 | 67.71 | 106.6 |
| anchor only | 28.417 | 21.015 | 17.235 | 0.9120 | 0.7828 | 0.5977 | 1.164 | 3.872 | 7.071 | 0.0998 | 0.2320 | 0.3875 | 18.67 | 56.52 | 102.5 |
| **Cross-Image Context** | | | | | | | | | | | | | | | |
| Methods | 0-20% | 20-40% | 40-60% | 0-20% | 20-40% | 40-60% | 0-20% | 20-40% | 40-60% | 0-20% | 20-40% | 40-60% | 0-20% | 20-40% | 40-60% |
| merged sets | 28.833 | 21.054 | 17.574 | 0.9130 | 0.7883 | 0.6035 | 1.145 | 3.721 | 6.857 | 0.0882 | 0.2217 | 0.3638 | 19.23 | 53.44 | 96.47 |
| **CICM** | **29.214** | **21.728** | **19.211** | **0.9205** | **0.8047** | **0.6258** | **1.079** | **3.478** | **6.375** | **0.0829** | **0.2045** | **0.3284** | **17.21** | **45.17** | **78.49** |

Table 2: The results of various ways of using context information on the test set of Places2.

RUC [4], which are the clustering methods for harnessing the single-image context in our scenario. In Figure 3, we show the process of using the clustering methods. For a single image, we use the clustering methods to divide the regional features of the complete regions into several clusters. Based on the similarities with the cluster centers, we find the most relevant regional features for augmenting the regional features of the corrupted regions. In the scenario of using the single-image context, we release the memory that stores different clusters of the regional features of each image, after using each mini-batch to optimize the network parameters.

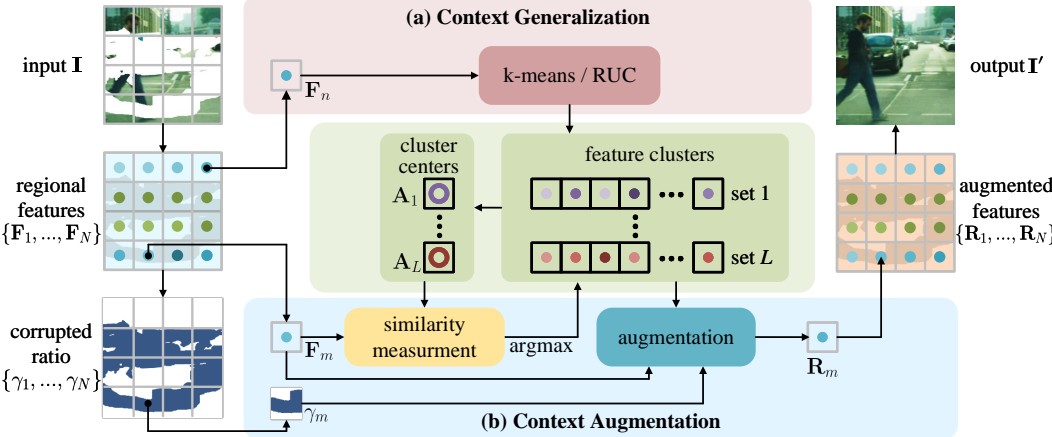

Figure 3: (a) For the regional features of the complete regions, the context generalization uses the clustering method (k-means or RUC) to cluster the regional features, where each cluster has a center. (b) Given a regional feature of the corrupted region, the context augmentation measures the similarity with every cluster center. It selects the cluster, where all regional features are used to augment the regional feature of the corrupted region.

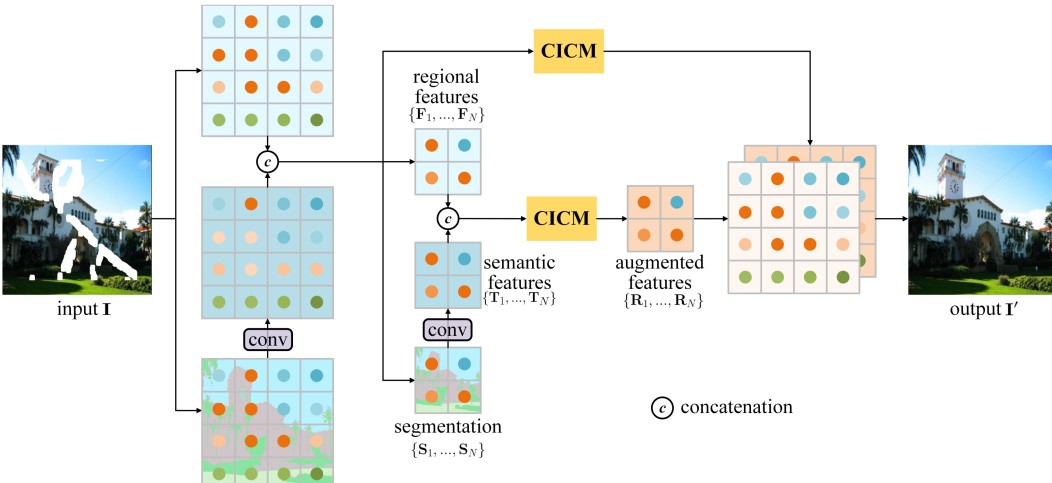

Figure 4: The extensive counterpart of our inpainting network with CICM. The network contains a lightweight segmentation sub-network for computing the semantic features.

**Extensive Evaluation on Semantic Inpainting** We extensively evaluate CICM on the public datasets (i.e., Cityscapes and Outdoor Scenes), which provide the semantic object categories for assisting the image inpainting task. Here, CICM can be easily extended to learn the cross-image features from not only the RGB images but also the semantic segmentation results. The results have been presented in Table 3 (also see Table 5 of the main paper).

In Figure 4, we provide more details of extending CICM. Along with the inpainting network, we train a lightweight semantic segmentation sub-network, which outputs the segmentation scores for all pixels in the input image. The segmentation scores are fed to a convolutional layer, which computes the semantic features. We concatenate the semantic features with the regional features. The concatenated features are used for computing the cross-image features in CICM.

| Cityscapes Dataset | | | | | | | | | | | | | | | |
|---|---|---|---|---|---|---|---|---|---|---|---|---|---|---|---|
| Methods | PSNR ↑ | | | SSIM ↑ | | | L1 ↓ | | | LPIPS ↓ | | | FID ↓ | | |
| | 0-20% | 20-40% | 40-60% | 0-20% | 20-40% | 40-60% | 0-20% | 20-40% | 40-60% | 0-20% | 20-40% | 40-60% | 0-20% | 20-40% | 40-60% |
| Gated [2] | 35.870 | 27.011 | 22.938 | 0.965 | 0.801 | 0.748 | 0.518 | 2.627 | 3.872 | 0.0345 | 0.0576 | 0.1573 | 5.298 | 23.28 | 52.29 |
| PEN [5] | 33.693 | 23.927 | 22.336 | 0.964 | 0.800 | 0.694 | 0.548 | 3.132 | 4.012 | 0.0317 | 0.0552 | 0.1662 | 8.314 | 48.67 | 66.70 |
| SPG [6] | 29.627 | 25.425 | 21.863 | 0.900 | 0.817 | 0.718 | 0.722 | 2.877 | 4.188 | 0.0412 | 0.0610 | 0.1627 | 17.14 | 27.09 | 36.42 |
| SWAP [7] | 32.973 | 26.112 | 22.984 | 0.965 | 0.885 | 0.782 | 0.602 | 2.435 | 3.762 | 0.0365 | 0.0547 | 0.1543 | 6.327 | 15.48 | 29.32 |
| SPL [8] | 35.543 | 27.639 | 23.530 | 0.969 | 0.892 | 0.773 | 0.476 | 2.192 | 3.353 | 0.0311 | 0.0486 | 0.1263 | 4.686 | 12.94 | 28.81 |
| **CICM** | **36.728** | **29.848** | **25.625** | **0.978** | **0.912** | **0.844** | **0.421** | **1.748** | **2.943** | **0.0281** | **0.0465** | **0.1170** | **3.437** | **8.246** | **12.16** |
| Outdoor Scenes Dataset | | | | | | | | | | | | | | | |
| Methods | PSNR ↑ | | | SSIM ↑ | | | L1 ↓ | | | LPIPS ↓ | | | FID ↓ | | |
| | 0-20% | 20-40% | 40-60% | 0-20% | 20-40% | 40-60% | 0-20% | 20-40% | 40-60% | 0-20% | 20-40% | 40-60% | 0-20% | 20-40% | 40-60% |
| Gated [2] | 30.826 | 24.262 | 19.294 | 0.955 | 0.874 | 0.680 | 0.910 | 2.018 | 2.684 | 0.0778 | 0.1578 | 0.2320 | 20.23 | 49.42 | 89.84 |
| PEN [5] | 29.072 | 21.515 | 19.237 | 0.949 | 0.796 | 0.630 | 1.036 | 2.438 | 2.711 | 0.0781 | 0.1662 | 0.2533 | 19.82 | 59.78 | 90.21 |
| SPG [6] | 24.156 | 21.692 | 18.282 | 0.801 | 0.685 | 0.533 | 1.417 | 2.638 | 2.764 | 0.0865 | 0.2218 | 0.2764 | 46.48 | 72.96 | 101.3 |
| SWAP [7] | 30.361 | 25.116 | 20.832 | 0.948 | 0.861 | 0.702 | 0.832 | 1.866 | 2.495 | 0.0572 | 0.1683 | 0.2217 | 13.29 | 40.01 | 63.86 |
| SPL [8] | 32.599 | 25.485 | 21.083 | 0.961 | 0.864 | 0.710 | 0.749 | 1.729 | 2.387 | 0.0465 | 0.1304 | 0.2042 | 11.24 | 30.07 | 53.28 |
| **CICM** | **33.271** | **26.467** | **22.116** | **0.969** | **0.886** | **0.732** | **0.674** | **1.411** | **2.011** | **0.0412** | **0.1141** | **0.1872** | **8.684** | **23.35** | **42.47** |

Table 3: Comparison with state-of-the-art methods on the test sets of Cityscapes and Outdoor Scenes.

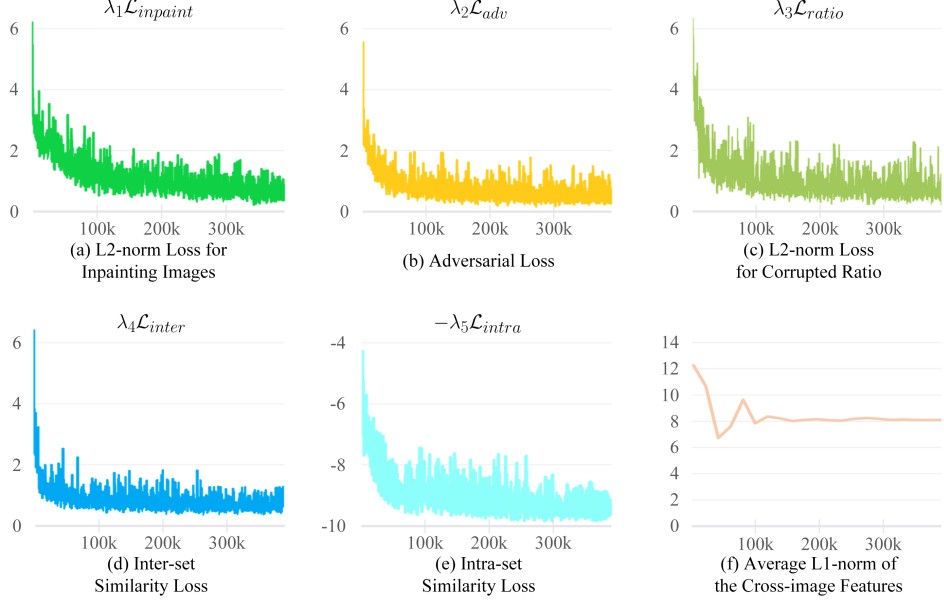

Figure 5: Analysis of convergence of the network training.

## 2.2 Analysis on Convergence of Network Training

To analyze the convergence of the network training, we show the changes of L2-norm and adversarial loss for penalizing the inpainting error, L2-norm for penalizing the estimation error of the corrupted ratios, inter-set and negative intra-set similarities, and average L1-norm of the cross-image features in CICM. The results are reported in Figure 5 (a–f), where the changes converge stalely at the final stages of the network training.

## 2.3 Analysis on Distributions of Cross-Image Features

We show the distributions of the cross-image features in CICM on different datasets in Figure 6. Here, we resort to t-SNE [9] for the visualization of the distributions of the cross-image features in the 2D space. During the network training, we also compute the weighted average of the image regions, whose regional features are injected by the context generalization into the cross-image features. We also show the average image regions in the corners of Figure 6. We find that most of the cross-image features, which belong to the same set, appear close to each other and represent similar visual patterns.

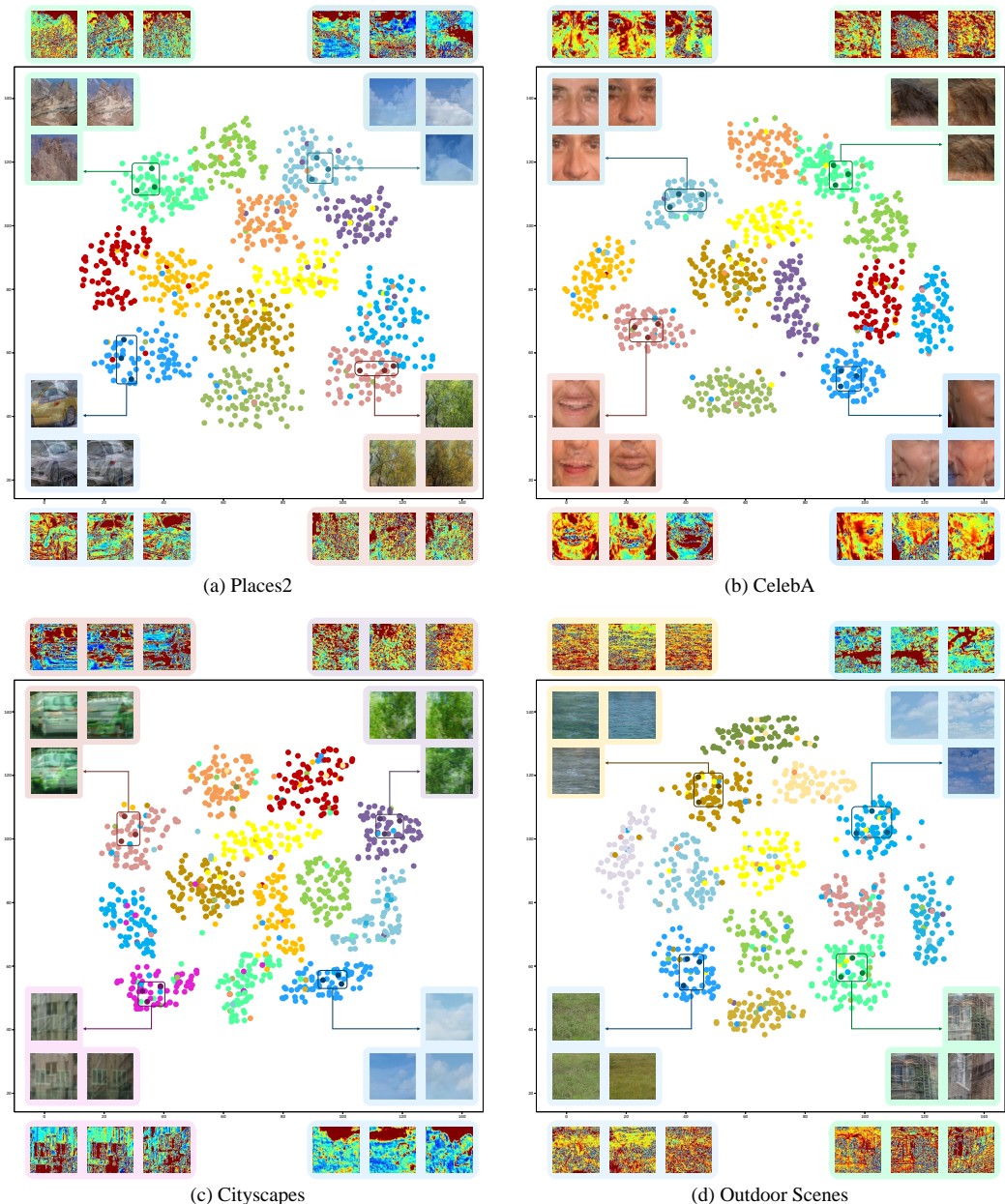

(a) Places2

(b) CelebA

(c) Cityscapes

(d) Outdoor Scenes

Figure 6: Distribution of the cross-image features in CICM on different datasets. A scatter point represents a cross-image feature, which is embedded into the 2D space. The scatter points with the same color represent the cross-image features in the same set of CICM.

## 2.4 More Visual Results

We provide more visual results on Places2, CelebA, Cityscapes and Outdoor Scenes in Figures 7, 8, 9, and 10. As shown in these visual results, our inpainting network with CICM generally produces the high-quality results.

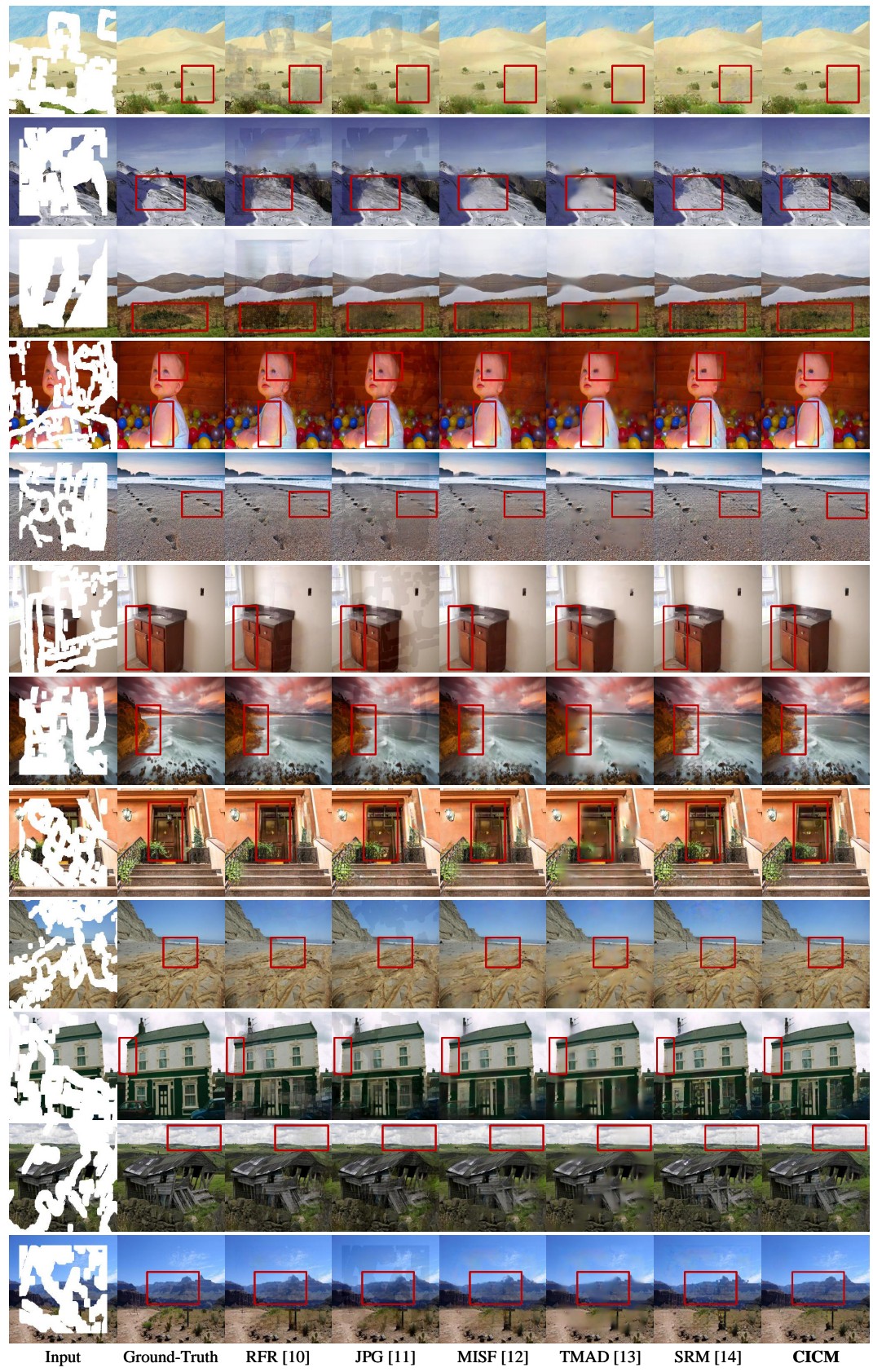

Input     Ground-Truth     RFR [10]     JPG [11]     MISF [12]     TMAD [13]     SRM [14]     **CICM**

Figure 7: Visual results of RFR [10], JPG [11], MISF [12], TMAD [13], SRM [14] and CICM on the test set of Places2.

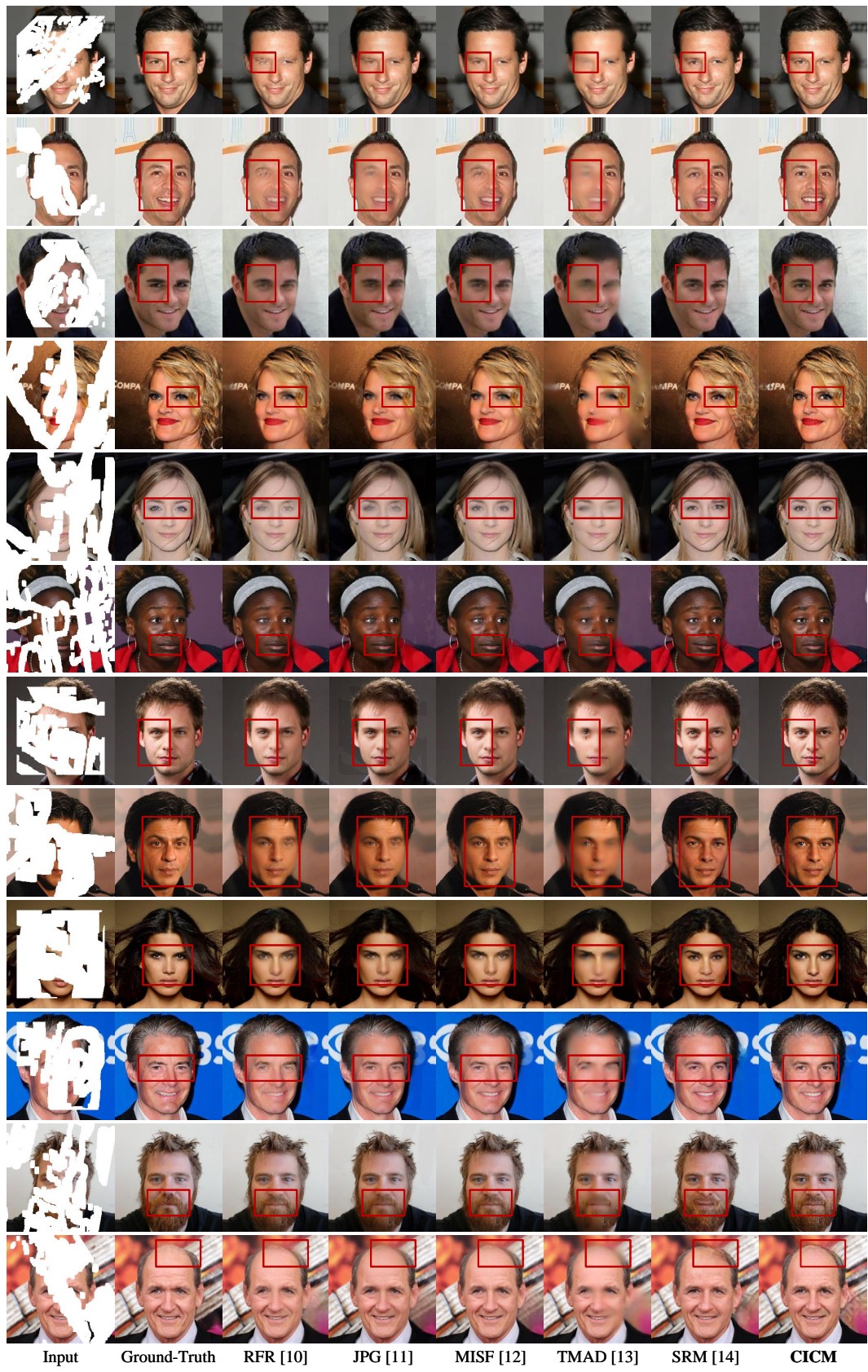

| Input | Ground-Truth | RFR [10] | JPG [11] | MISF [12] | TMAD [13] | SRM [14] | **CICM** |

Figure 8: Visual results of RFR [10], JPG [11], MISF [12], TMAD [13], SRM [14] and CICM on the test set of CelebA.

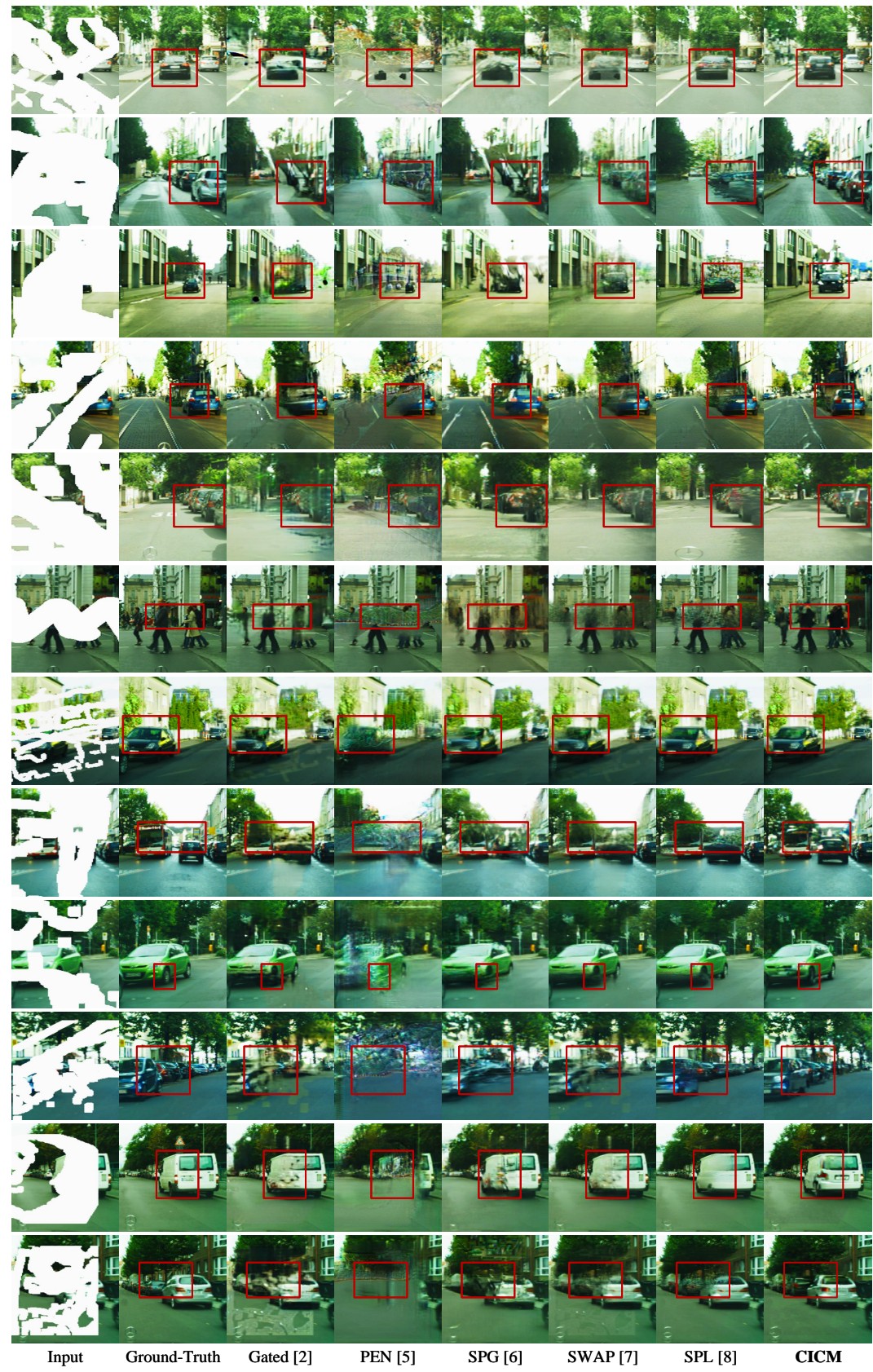

Input  Ground-Truth  Gated [2]  PEN [5]  SPG [6]  SWAP [7]  SPL [8]  **CICM**

Figure 9: Visual results of Gated [2], PEN [5], SPG [6], SWAP [7], SPL [8] and CICM on the test set of Cityscapes.

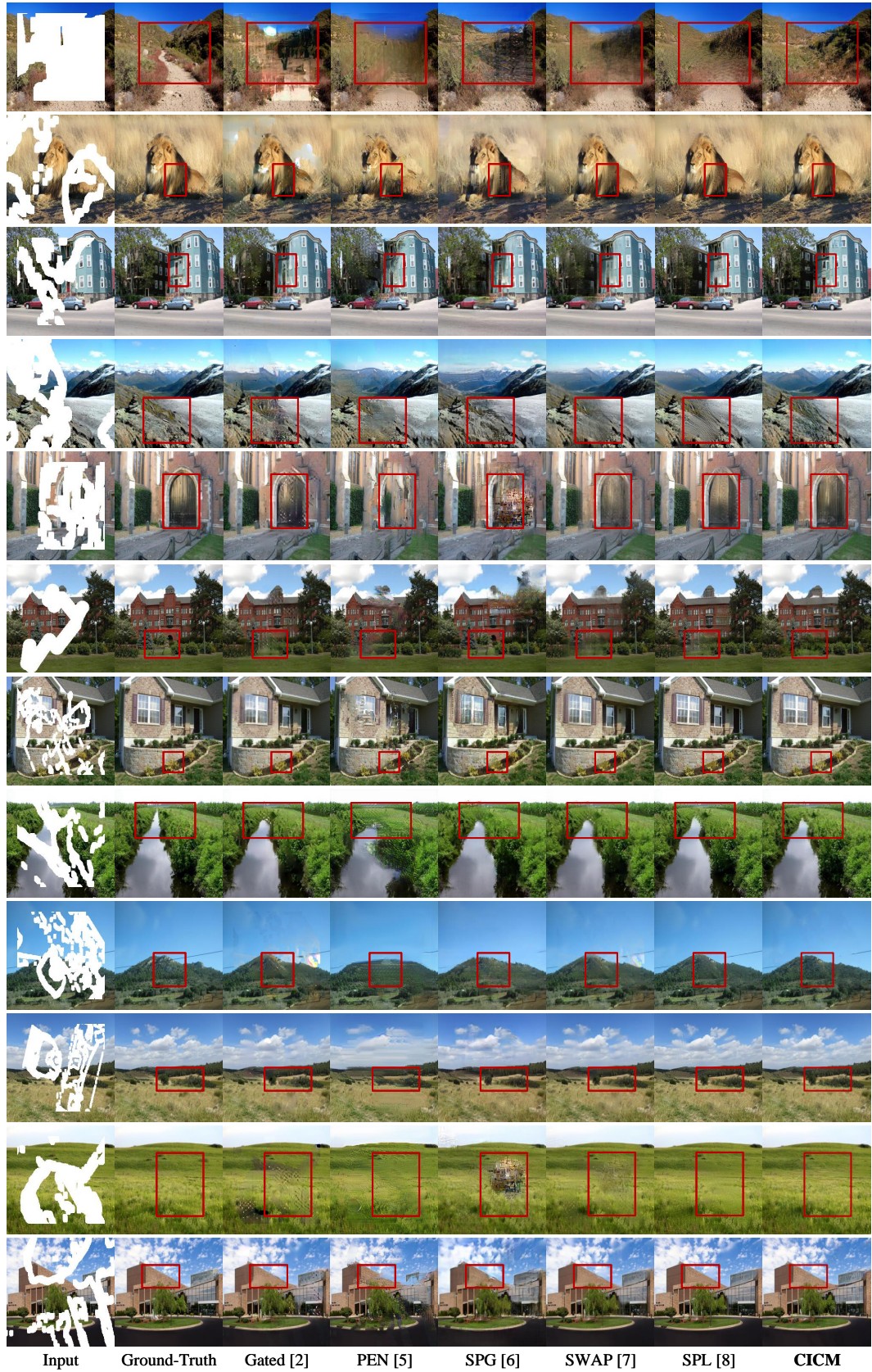

Input     Ground-Truth     Gated [2]     PEN [5]     SPG [6]     SWAP [7]     SPL [8]     **CICM**

Figure 10: Visual results of Gated [2], PEN [5], SPG [6], SWAP [7], SPL [8] and CICM on the test set of Outdoor Scenes.

# 3 Limitation

## 3.1 Failure Cases

Some failure cases are shown in Figure 11 for better understanding the limitation of our method. In these cases, the input images have large scopes of the corrupted regions (see (a) and (b)). The input images provide little information for the context augmentation, disallowing the context augmentation to reliably find the relevant cross-image features in CICM, consequently offering less useful context for recovering the corrupted regions.

In some of these failure cases, the input images contains the visual information, which shows a large discrepancy with the information learned and stored in CICM. For example, the faces in Figure 11(c–d) are observed from the angles that are rarely seen in the training data. Though CICM contains the cross-image features produced by the context generalization, these challenging cases still leads to unsatisfactory results. Thus, the generalization power of CICM still need to be improved.

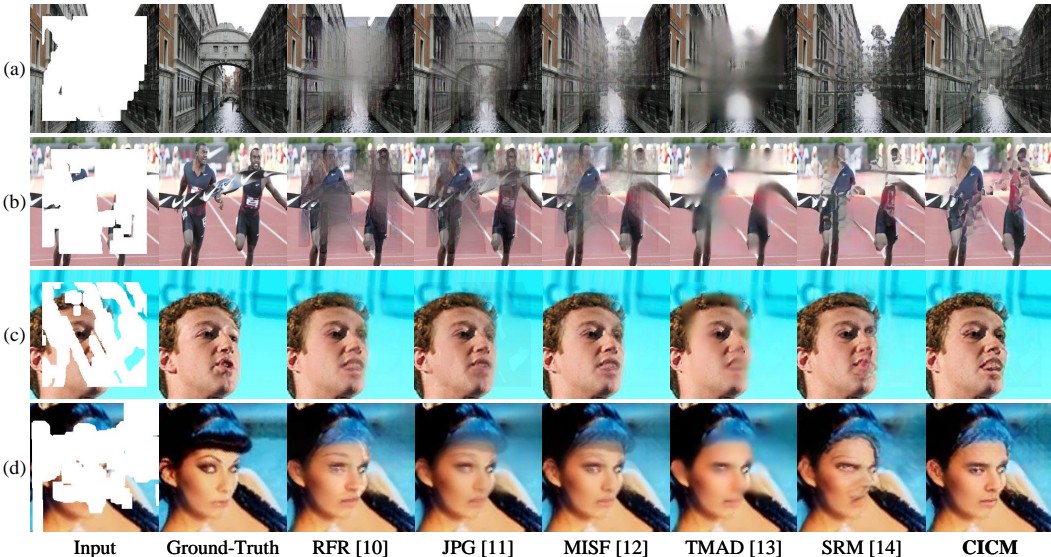

Figure 11: The visual results failure cases on the test set of Places2 and CelebA.

## 3.2 Memory Increase

In Table 4 (also see Table 3 of the main paper), we have justify the generalization power of CICM, which consistently improves the performances of different inpainting networks. It should be noted that CICM requires more memory budget for storing the cross-image features. In Table 5, we compare the network parameters (M), GPU memory (GB), and FLOPs (G) of the inpainting networks with and without CICM, for considering the trade-off between the inpainting performance and computational efficiency.

## 3.3 Evaluation of CICM in Cross-Model and -Dataset Scenarios

Note that CICM can be added to different inpainting networks. Here, we evaluate the performance of CICM, which is trained along with an inpainting networks and applied to another network. We report the results in Table 6. Here, we evaluate the inpainting performance on Place2, where the corrupted ratio is set to 20-40%.

We train the baseline UNet and the recent inpainting method MISF [12], which are equipped with CICMs respectively. Their performances are reported in the row "w/o Cross Model". Then, we exchange CICMs between UNet and MISF, where each of these CICMs are directly used for inpainting without further fine-tuning. The performances of UNet and MISF with the exchanged CICMs are reported in the row "w/ Cross Model". We find that the exchanged CICMs slightly degrade the performances of UNet and MISF. It may be because the cross-image features in the exchanged CICMs mismatch the regional features extracted by UNet and MISF. Yet, the exchanged CICMs yield better performances than the networks without CICM (see the row "w/o CICM").

| Methods | PSNR ↑ | | | SSIM ↑ | | | L1 ↓ | | | LPIPS ↓ | | | FID ↓ | | |
|---|---|---|---|---|---|---|---|---|---|---|---|---|---|---|---|
| | 0-20% | 20-40% | 40-60% | 0-20% | 20-40% | 40-60% | 0-20% | 20-40% | 40-60% | 0-20% | 20-40% | 40-60% | 0-20% | 20-40% | 40-60% |
| | | | | | | Places2 Dataset | | | | | | | | | |
| UNet | 28.637 | 20.944 | 17.022 | 0.9141 | 0.7885 | 0.5746 | 1.137 | 3.606 | 7.269 | 0.0850 | 0.2162 | 0.3838 | 18.37 | 58.22 | 112.7 |
| **UNet-CICM** | **29.214** | **21.728** | **18.811** | **0.9205** | **0.8047** | **0.6258** | **1.079** | **3.478** | **6.375** | **0.0829** | **0.2045** | **0.3284** | **17.21** | **45.17** | **78.49** |
| RFR [10] | 28.891 | 21.278 | 17.648 | 0.9167 | 0.7893 | 0.5953 | 1.128 | 3.532 | 6.916 | 0.0873 | 0.2267 | 0.3723 | 17.83 | 51.29 | 95.72 |
| **RFR-CICM** | **29.411** | **22.146** | **19.313** | **0.9210** | **0.8134** | **0.6311** | **1.065** | **3.337** | **6.211** | **0.0834** | **0.2088** | **0.3174** | **16.69** | **40.23** | **64.17** |
| JPG [11] | 30.023 | 22.561 | 18.045 | 0.9362 | 0.8267 | 0.6762 | 0.902 | 2.671 | 5.725 | 0.0883 | 0.2417 | 0.3521 | 16.78 | 39.21 | 78.77 |
| **JPG-CICM** | **30.457** | **23.716** | **20.016** | **0.9417** | **0.8325** | **0.7022** | **0.868** | **2.516** | **5.073** | **0.0835** | **0.2174** | **0.3093** | **16.02** | **34.88** | **58.19** |
| MISF [12] | 31.044 | 23.799 | 19.314 | 0.9443 | 0.8312 | 0.6736 | 0.741 | 2.520 | 5.311 | 0.0537 | 0.1721 | 0.2821 | 16.39 | 35.31 | 62.67 |
| **MISF-CICM** | **31.516** | **24.858** | **21.267** | **0.9491** | **0.8405** | **0.7027** | **0.712** | **2.317** | **4.872** | **0.0501** | **0.1498** | **0.2389** | **14.76** | **29.12** | **48.21** |
| | | | | | | CelebA Dataset | | | | | | | | | |
| UNet | 33.133 | 24.573 | 19.522 | 0.9577 | 0.8621 | 0.7234 | 0.533 | 1.882 | 4.623 | 0.0432 | 0.1282 | 0.2476 | 10.74 | 40.83 | 75.39 |
| **UNet-CICM** | **33.388** | **25.384** | **21.673** | **0.9610** | **0.8788** | **0.7689** | **0.518** | **1.820** | **4.127** | **0.0419** | **0.1214** | **0.2238** | **8.493** | **35.47** | **63.22** |
| RFR [10] | 33.327 | 25.224 | 20.133 | 0.9571 | 0.8722 | 0.7323 | 0.538 | 1.872 | 4.638 | 0.0437 | 0.1257 | 0.2421 | 9.362 | 33.28 | 67.31 |
| **RFR-CICM** | **33.636** | **26.056** | **21.517** | **0.9601** | **0.8793** | **0.7654** | **0.515** | **1.784** | **4.164** | **0.0408** | **0.1166** | **0.2287** | **7.134** | **31.78** | **56.98** |
| JPG [11] | 33.925 | 26.338 | 20.548 | 0.9573 | 0.8826 | 0.7428 | 0.527 | 1.692 | 4.411 | 0.0427 | 0.1307 | 0.2559 | 8.273 | 32.02 | 61.32 |
| **JPG-CICM** | **34.262** | **27.027** | **22.393** | **0.9619** | **0.8902** | **0.7681** | **0.504** | **1.646** | **3.817** | **0.0401** | **0.1186** | **0.2265** | **6.374** | **29.26** | **53.87** |
| MISF [12] | 34.302 | 26.387 | 21.289 | 0.9629 | 0.8903 | 0.7585 | 0.501 | 1.572 | 3.922 | 0.0336 | 0.0981 | 0.2137 | 6.836 | 30.11 | 55.75 |
| **MISF-CICM** | **34.695** | **27.854** | **23.338** | **0.9683** | **0.9012** | **0.7782** | **0.489** | **1.502** | **3.311** | **0.0317** | **0.0925** | **0.1921** | **5.023** | **27.99** | **47.12** |

Table 4: The results of combining CICM with different inpainting networks (i.e., RFR [10], JPG [11], and MISF [12]) on the test sets of Places2 and CelebA.

| Methods | Parameters (M) | Memory (GB) | FLOPs (G) | Methods | Parameters (M) | Memory (GB) | FLOPs (G) |
|---|---|---|---|---|---|---|---|
| UNet | 10.42 | 11.37 | 10.02 | JPG | 42.57 | 39.22 | 31.79 |
| **UNet-CICM** | **11.21** | **13.48** | **11.75** | **JPG-CICM** | **44.21** | **42.27** | **33.51** |
| RFR | 19.58 | 24.98 | 27.72 | MISF | 37.21 | 36.74 | 15.26 |
| **RFR-CICM** | **20.33** | **27.33** | **29.11** | **MISF-CICM** | **38.43** | **38.15** | **17.20** |

Table 5: Comparison of the network parameters (M), GPU memory (GB), and FLOPs (G) of the inpainting networks with and without CICM.

| | PSNR ↑ | SSIM ↑ | L1 ↓ | LPIPS ↓ | FID ↓ | PSNR ↑ | SSIM ↑ | L1 ↓ | LPIPS ↓ | FID ↓ |
|---|---|---|---|---|---|---|---|---|---|---|
| w/o CICM | | UNet | | | | | MISF | | | |
| | 20.944 | 0.7885 | 3.606 | 0.2162 | 57.38 | 23.799 | 0.8312 | 2.522 | 0.1721 | 34.72 |
| w/o Cross Model | | UNet-CICM | | | | | MISF-CICM | | | |
| | **21.728** | **0.8047** | **3.478** | **0.2045** | **45.17** | **24.858** | **0.8405** | **2.317** | **0.1498** | **29.12** |
| w/ Cross Model | | UNet-CICM (MISF) | | | | | MISF-CICM (UNet) | | | |
| | 21.373 | 0.8001 | 3.554 | 0.2127 | 48.27 | 24.235 | 0.8335 | 2.422 | 0.1574 | 32.35 |

Table 6: The results of the methods replacing the CICMs of UNet-CICM and MISF-CICM on the test sets of Places2.

In Table 7, we investigate the possibility of exchanging CICMs that are trained on different datasets. Here, we train two separate UNets on Places2 and CelebA. Each UNet is associated with CICM.

| | Places2 | | | | | CelebA | | | | |
|---|---|---|---|---|---|---|---|---|---|---|
| | PSNR ↑ | SSIM ↑ | L1 ↓ | LPIPS ↓ | FID ↓ | PSNR ↑ | SSIM ↑ | L1 ↓ | LPIPS ↓ | FID ↓ |
| w/o CICM | 20.944 | 0.7885 | 3.606 | 0.2162 | 57.38 | 24.573 | 0.8621 | 1.882 | 0.1282 | 40.27 |
| w/o Cross Dataset | **21.728** | **0.8047** | **3.478** | **0.2045** | **45.17** | **25.384** | **0.8788** | **1.820** | **0.1214** | **35.47** |
| w/ Cross Dataset | 19.274 | 0.7672 | 3.936 | 0.2237 | 75.32 | 23.477 | 0.8489 | 2.024 | 0.1473 | 67.21 |

Table 7: The results of the methods replacing the CICMs of two UNet-CICM trained on Places and CelebA respectively.

After the network training, we exchange CICMs of the two UNets, which are evaluated on the test sets of Places2 and CelebA respectively (see the row "w/ Cross Dataset"). We find that the exchanged CICMs drastically degrade the performances, compared to the inpainting networks without the exchanged CICMs (see the row "w/o Cross Dataset") or even without CICM (see the row "w/o CICM"). This may because in Places2 and CelebA, the images contains scene and face information, respectively, showing a weak correlation. Thus, the cross-image features in the exchanged CICMs likely mislead the context augmentation.

## 4   Code Segment

Our code will be available at: https://github.com/fengtl/CICM.