# OpenReview forum: "Cross-Image Context for Single Image Inpainting"
_NeurIPS.cc/2022/Conference — NeurIPS 2022 Accept_

### Official Review · Reviewer_3YLw · 2022-07-10

**Rating:** 6
**Confidence:** 4
**Soundness:** 3 good
**Presentation:** 3 good
**Contribution:** 3 good

**Summary:**

This paper proposes the Cross-Image Context Memory (CICM) for learning and using the cross-image context to recover corrupted images. CICM consists of multiple sets of cross-image features learned from the image regions with different visual patterns. The regional features are learned across different images, thus providing richer context that benefits the inpainting task. The experimental results demonstrate the effectiveness and generalization of CICM, which achieves state-of-the-art performances on various datasets for single image inpainting.

**Questions:**

1)	Five losses are used to train the model. Is there any balance for their contributions? Since there are different terms in each loss, why are the magnitude of these loss values the same as in Fig. 5 in the supplementary?
2)	How are the anchor features and feature sets initialized?


**Ethics Review Area:**

["Privacy and Security (e.g., consent)"]

**Strengths And Weaknesses:**

Strengths
1）	The utilization of the cross-image context to assist in image inpainting is reasonable and the proposed cross-image context memory is somewhat novel and can also be generalized to the existing inpainting models.
2）	The experiments are sufficient, and the internal study is nice to show the effectiveness of the CICM
3）	The presentation is clear, and the reference is also adequate.

Weaknesses
1)	Besides the number of feature sets and the size of each set, the resolution of the regional features and the number of layers adopting the CICM are also vital settings that affect the image inpainting performance, but there seems no explanation and analysis for them.

Minor:
1)	In line 74, symbols of H and W are used to denote the height and width of the image and the feature, but they are actually different.
2)	In line 117, what is the value of the momentum factor.
3)	In line 168, “four groups” is “three groups”.

---

> ### Author Response · Authors · 2022-08-01
> **Response to Reviewer 3YLw**
>
> **1. The resolution of the regional features and the number of layers adopting the CICM are also vital settings that affect the image inpainting performance.**
>
> We are sorry for missing these results. We have experimented with changing the resolution of regional features and the number of layers adopting the CICM on Places2 dataset by using a UNet with 15 convolutional layers as the backbone. We provide these results in the two tables below.
>
> ||||||||Table 1|||||||||
> |:-:|:-:|:-:|:-:|:-:|:-:|:-:|:-:|:-:|:-:|:-:|:-:|:-:|:-:|:-:|:-:|
> |||PSNR↑|||SSIM↑|||L1↓|||LPIPS↓|||FID↓||
> ||0-20%|20-40%|40-60%|0-20%|20-40%|40-60%|0-20%|20-40%|40-60%|0-20%|20-40%|40-60%|0-20%|20-40%|40-60%|
> |2×2|28.92|21.04|18.89|0.913|0.794|0.618|1.104|3.589|6.817|0.0883|0.2210|0.3412|19.01|52.38|86.39|
> |4×4|29.11|21.48|19.07|0.917|0.798|0.620|1.098|3.532|6.541|0.0845|0.2137|0.3349|18.31|48.38|83.11|
> |8×8|29.21|21.73|19.21|0.921|0.805|0.626|1.079|3.478|6.375|0.0829|0.2045|0.3284|17.21|45.17|78.49|
> |16×16|29.14|21.66|19.08|0.920|0.804|0.622|1.083|3.482|6.488|0.0841|0.2075|0.3331|17.95|47.28|82.69|
>
> ||||||||Table 2|||||||||
> |:-:|:-:|:-:|:-:|:-:|:-:|:-:|:-:|:-:|:-:|:-:|:-:|:-:|:-:|:-:|:-:|
> |||PSNR↑|||SSIM↑|||L1↓|||LPIPS↓|||FID↓||
> ||0-20%|20-40%|40-60%|0-20%|20-40%|40-60%|0-20%|20-40%|40-60%|0-20%|20-40%|40-60%|0-20%|20-40%|40-60%|
> |1 layer|29.21|21.73|19.21|0.921|0.805|0.626|1.079|3.478|6.375|0.0829|0.2045|0.3284|17.21|45.17|78.49|
> |2 layers|29.74|22.00|19.72|0.927|0.809|0.651|0.993|3.385|5.834|0.0859|0.1972|0.3175|16.98|43.28|67.39|
> |3 layers|30.02|22.57|20.21|0.934|0.816|0.677|0.921|3.321|5.516|0.0885|0.1880|0.3086|16.62|39.02|57.62|
> |4 layers|30.03|22.68|20.55|0.935|0.815|0.689|0.902|3.315|5.370|0.0881|0.1853|0.3011|16.48|37.59|55.37|
>
> In Table 1, we select the resolution of regional features from the set {2x2, 4x4, 8x8, and 16x16} and report the performances. Here, we use the backbone UNet to output the convolutional map with the lowest resolution. By subdividing the convolutional map into a set of 8x8 regional features, we achieve the best results in Table 1.
>
> In Table 2, we equip CICMs to different convolutional layers (1,2,3,4) and report the results. Here, 1 means the deepest layer that outputs the convolutional feature map with the lowest resolution. In each case, we always equip the deeper layers with CICMs. By adding CICMs to more layers, we achieve better results. We keep using CICM at 4 layers in Tables 4 and 5 in the paper.
>
> **2. In line 74, symbols of H and W are used to denote the height and width of the image and the feature, but they are different. In line 117, what is the value of the momentum factor? In line 168, “four groups” is “three groups”.**
>
> Thanks for your correction. In line 74, we let the height and width of the image and the feature be the same, to simplify the notations in this paper. In line 117, the momentum factor is 0.5. In line 169, we have corrected “four groups” to “three groups”.
>
> **3. Five losses are used to train the model. Is there any balance for their contributions? Since there are different terms in each loss, why are the magnitude of these loss values the same as in Fig. 5 in the supplementary?**
>
> Thanks. Different balance coefficients are used for the losses in Eq. 9 (L_inpaint 1.0, L_adv 0.1, L_ratio 1.0, L_inter 20, and L_intra 0.5). In Figure 5 of the supplementary file, we have multiplied the balance coefficients by the corresponding losses, which thus have similar magnitudes in the figure.
>
> **4. How are the anchor features and feature sets initialized?**
>
> In our implementation, we use a warm-up strategy to pre-train the backbone network for 50K iterations. The encoder of the pre-trained backbone is used to compute the regional features of different images. We conduct k-means clustering on the regional features, computing the cluster centers as the initial anchor features. The regional features, which are nearest to the initial anchor features, are selected as the initial cross-image features in different sets of CICM. We will add this detail to the supplementary file.

---

> ### Author Response · Authors · 2022-08-07
> **Response to Reviewer 3YLw**
>
> Dear Reviewer 3YLw,
>
> We thank you again for your valuable comments, which significantly help us to polish our paper. We are looking to discussing with you the questions that are addressed unsatisfactorily.
>
> Best,
>
> Authors of Paper ID 465

---

> ### Author Response · Authors · 2022-08-08
> **Sincerely Request Your New Comment**
>
> Dear Reviewer 3YLw,
>
> Thanks for your review again. As the deadline for the authors' response is approaching, we sincerely request your comment on our primary response. This will definitely give us a valuable chance to address the questions unsolved.
>
> Best,
>
> Authors of Paper ID 465

---

### Official Review · Reviewer_GGCD · 2022-07-11

**Rating:** 8
**Confidence:** 4
**Soundness:** 4 excellent
**Presentation:** 4 excellent
**Contribution:** 4 excellent

**Summary:**

This paper proposes an image inpainting algorithm that learns visual context features across different images and saves them in an external memory (CICM). These features are used to augment regional features of the corrupted input image, which may result in better completion quality than relying on features inside the single image. The proposed approach outperforms existing models on the public datasets.

**Questions:**

How is the CICM initialized?

What is the backbone network of the proposed model in Tables 4 and 5?

Would the quality enhancement by applying CICM scheme be outstanding enough compared to the complexity increase (i.e., trade-off)?

The search into the CICM is not a bottleneck of the model performance, which is not addressed?

**Limitations:**

Yes, they addressed the limitations in Sections 3 and 4 of the supplementary materials, which include failure cases, memory increase, cross model and cross dataset scenarios, and societal impacts. These discussions seem to have been thoroughly made.

**Strengths And Weaknesses:**

* Strengths:
  - The proposed approach based on cross-image context memory is novel. It saves higher level richer visual context thus is unlike previous memory based methods such as TMAD [42] and SRM [43].
  - The proposed approach is highly effective, outperforming recent existing works (Tables 4 and 5).
  - The proposed approach is flexible and general, capable of extending several existing frameworks with consistent enhancements (Table 3).
  - Ablation studies on internal component of the proposed approach have been thoroughly made thus proved their importance (Tables 1 and 2).
  - Several extensions and variants of the proposed approach have been explored in the supplementary materials, all showing meaningful results.

* Weaknesses:
  - It is unclear how the CICM is initialized at the beginning of the training.
  - It is unclear which architecture is used as their default backbone network.
  - The use of an external memory bank increases the ram usage, parameters and FLOPs quite a bit.
  - Depending on the scale of CICM and device property, the searching processes may become a bottleneck of the inference speed.


----- Comments after reading the rebuttal -----

I believe that the proposed work describes a very interesting algorithm based on CICM, yet unseen from existing works. As addressed by the authors, the CICM is fast to search into and effective to produce higher quality generation by utilizing compact cross-image information unavailable within the input image. The quality gains are quite clear without much sacrificing the efficiency. My final recommendation is still to strongly support to accept the paper.

---

> ### Author Response · Authors · 2022-08-01
> **Response to Reviewer GGCD**
>
> **1. How is the CICM initialized?**
>
> In our implementation, we use a warm-up strategy to pre-train the backbone network for 50K iterations. The encoder of the pre-trained backbone is used to compute the regional features of different images. We conduct k-means clustering on the regional features, computing the cluster centers as the initial anchor features. The regional features, which are nearest to the initial anchor features, are selected as the initial cross-image features in different sets of CICM. We will add this detail to the supplementary file.
>
> **2. What is the backbone network of the proposed model in Tables 4 and 5?**
>
> In Tables 4 and 5, we use a UNet with 15 convolutional layers as the backbone. We have clarified this in the revised paper (see lines 254-255).
>
> **3. Would the quality enhancement by applying the CICM scheme be outstanding enough compared to the complexity increase (i.e., trade-off)?**
>
> Thanks for your comment. The complexity increase of CICM has been shown in Table 5 of the supplementary file (at most 2M Parameters, 3GB Memory, 2G FLOPs). At these costs, we achieve very consistent performance gains by using CICM (up to 1.845 PSNR, 0.0355 SSIM, 0.672 L1, 0.0491 LPIPS, and 25.20 FID on Places2; 1.857 PSNR, 0.0309 SSIM, 0.544 L1, 0.0221 LPIPS and 9.645 FID on CelebA). Please also see Table 3 of the paper for the performance gains. We have discussed the trade-off between performance and efficiency in Section 3.2 of the supplementary file. One may use this discussion to consider the trade-off between performance and efficiency.
>
> **4. The search into CICM is a bottleneck of the model performance.**
>
> Please note that the search into CICM is fast. First, we use the anchor features to select a feature set (see Eq. 4). Next, we use all of the cross-image features in the selected set to augment the regional feature, where the augmentation can be implemented as the matrix multiplication and accelerated by GPU. In our implementation, the search into CICM only occupies about 2% of the testing time.

---

> ### Author Response · Authors · 2022-08-07
> **Response to Reviewer GGCD**
>
> Dear Reviewer GGCD,
>
> We thank you again for your valuable comments, which significantly help us to polish our paper. We are looking to discussing with you the questions that are addressed unsatisfactorily.
>
> Best,
>
> Authors of Paper ID 465

---

> ### Comment · Reviewer_GGCD · 2022-08-08
> **Comments after reading the rebuttal**
>
> I believe that the proposed work describes a very interesting algorithm based on CICM, yet unseen from existing works. As addressed by the authors, the CICM is fast to search into and effective to produce higher quality generation by utilizing compact cross-image information unavailable within the input image. The quality gains are quite clear without much sacrificing the efficiency. My final recommendation is still to strongly support to accept the paper.

---

> > ### Author Response · Authors · 2022-08-08
> > **Thanks for Your Review**
> >
> > Dear Reviewer GGCD,
> >
> > Thank you again for your review. We are pleased to see that the questions raised by you are solved.
> >
> > Best,
> >
> > Authors of Paper ID 465

---

### Official Review · Reviewer_8FQc · 2022-07-11

**Rating:** 5
**Confidence:** 4
**Soundness:** 3 good
**Presentation:** 3 good
**Contribution:** 3 good

**Summary:**

The paper proposes to use the cross-image context which consists of features learned from different visual patterns to recover the corrupted regions. The proposed method archives state-of-the-art performance on single-image inpainting on multiple datasets.

**Questions:**

Please address my concerns shown in the weaknesses.

**Ethics Review Area:**

["I don’t know"]

**Strengths And Weaknesses:**

Pros:
The proposed CICM achieves stable improvements on all the baseline methods and achieves state-of-art inpainting performance.
The authors conduct extensive experiments for the ablation study to validate the design of context generalization and context augmentation. The design of these modules is validated.

Cons:
The core contribution seems not strong enough for me.  As the external memory bank-based method has been adopted in many tasks, what is the unique design to make it suitable for inpainting? On the other hand, the visual quality improvement is also not obvious enough. Even with the proposed CICM, the inpainting results are not as attractive as LaMa and RePaint.

The writing needs to be improved. There are many sentences that are not meaningful but very complicated (e.g., lines 36 to 39).

How is the external memory bank-based method compared to gan prior-based methods such as LAMA?
Is it possible to visualize the features learned in CICM? It is interesting to see how the CICM is learned during the training.

Suggestions:
Missing reference in Figure 4.

---

> ### Author Response · Authors · 2022-08-01
> **Response to Reviewer 8FQc**
>
> **1. The core contribution seems not strong enough. What is the unique design to make CICM suitable for inpainting?**
>
> Our method of learning the cross-image context in CICM is non-trivial. The existing methods use the single-image context or class-specific information to recover the corrupted images, which however lacks visual information for computing the context. CICM stores the cross-image features learned from different images. It takes the advantage of more useful regions across images, providing richer context for recovering a region. CICM allows the inpainting to benefit from not only a kind of specific context. As discussed in Section 5.2 “Extensive Evaluation on Semantic Inpainting” of the paper, CICM stores the cross-image context learned from RGB images and segmentation results, which improve the results on the semantic inpainting task.
>
> Our contribution not only lies in the methodology ground but also in the extensive thinking, evaluation, and discussion of the CICM. CICM is a separable component alongside the inpainting network. Is it possible to train CICM with a network and apply it to another network? Is it also possible to train CICM on a dataset and apply it to the inpainting on another dataset? These questions are highly correlated to the generalization of CICM, but they are answered by few literatures. In Section 3.3 of the supplementary file, we answer these questions by evaluating CICM in the cross-model and -dataset scenarios. The cross-model CICM achieves better results than the methods without CICM. It demonstrates the capacity of CICM for transferring the learned context between different models. Yet, CICM degrades the performances in the cross-dataset scenario. We have provided our explanation of the degradation, pointing out a direction for improving the generalization of cross-image context.
>
> **2. The visual quality improvement is not obvious. The results are not as attractive as LaMa and RePaint.**
>
> More visualization results are provided in the supplementary file. Please zoom in on the results for a better visual quality.
>
> CICM is a component with generalized cross-image context for assisting different inpainting networks, rather than a stand-alone network that surpasses all of the existing networks. The generalization of CICM has been discussed in Section 5.2 “Combination with Different Inpainting Networks”. We compare the latest networks (RFR. JPG, and MISF) with/without CICM. CICM consistently improve these networks on Places2 and CelebA datasets.
>
> We also compare LaMa and RePaint with/without CICM. We use the pre-trained parameters and fine-tune the networks with CICM. Again, CICM helps LaMa and RePaint to achieve better results. Please see the results below.
> ||||||||Places2|||||||||
> |:-:|:-:|:-:|:-:|:-:|:-:|:-:|:-:|:-:|:-:|:-:|:-:|:-:|:-:|:-:|:-:|
> |||PSNR↑|||SSIM↑|||L1↓|||LPIPS↓|||FID↓||
> ||0-20%|20-40%|40-60%|0-20%|20-40%|40-60%|0-20%|20-40%|40-60%|0-20%|20-40%|40-60%|0-20%|20-40%|40-60%|
> |LaMa|31.64|24.87|21.13|0.952|0.846|0.701|0.742|2.375|4.925|0.0424|0.1275|0.2227|16.32|33.48|63.87|
> |LaMa-CICM|31.72|25.94|22.67|0.956|0.859|0.719|0.711|2.242|4.072|0.0398|0.1113|0.1884|14.22|29.94|55.49|
> |RePaint|31.75|24.97|21.35|0.953|0.848|0.708|0.725|2.223|4.873|0.0411|0.1241|0.2153|14.49|29.84|58.82|
> |RePaint-CICM|31.88|26.21|22.93|0.959|0.862|0.722|0.686|2.098|4.002|0.0386|0.1089|0.1780|11.57|25.39|51.16|
>
> ||||||||CelebA|||||||||
> |:-:|:-:|:-:|:-:|:-:|:-:|:-:|:-:|:-:|:-:|:-:|:-:|:-:|:-:|:-:|:-:|
> |||PSNR↑|||SSIM↑|||L1↓|||LPIPS↓|||FID↓||
> ||0-20%|20-40%|40-60%|0-20%|20-40%|40-60%|0-20%|20-40%|40-60%|0-20%|20-40%|40-60%|0-20%|20-40%|40-60%|
> |LaMa|34.57|26.79|21.97|0.971|0.893|0.772|0.479|1.582|3.774|0.0313|0.0924|0.1846|5.539|21.15|53.19|
> |LaMa-CICM|34.69|27.93|23.22|0.975|0.904|0.798|0.441|1.335|3.177|0.0289|0.0804|0.1602|4.127|17.92|41.87|
> |RePaint|34.57|26.88|22.15|0.972|0.898|0.778|0.472|1.563|3.573|0.0302|0.0912|0.1802|4.370|20.03|48.20|
> |RePaint-CICM|34.72|28.02|23.69|0.979|0.907|0.802|0.428|1.307|2.963|0.0275|0.0785|0.1577|3.237|16.63|35.54|
>
> **3. There are many sentences that are not meaningful but very complicated.**
>
> Thanks. We have rephrased the complicated in the revised paper.
>
> **4. Visualize the features learned in CICM.**
>
> In Figure 6 of the supplementary file, we have visualized the distribution of the cross-image features in different/identical set(s) of CICM. We use t-SNE to map these features into a 2D latent space for visualization. Different/identical set(s) of the cross-image features appear closely/far. It means that they have a large diversity/consistence. The diversity in different feature sets provides richer context for inpainting. The consistence in the identical set reduces the unreasonable cases, where discrepant contents are predicted for the similar regions. We will add the visualized feature maps.
>
> **5. Missing reference in Figure 4.**
>
> Thanks. We have added the references to Figure 4 in the revised paper.

---

> ### Author Response · Authors · 2022-08-07
> **Response to Reviewer 8FQc**
>
> Dear Reviewer 8FQc,
>
> We thank you again for your valuable comments, which significantly help us to polish our paper. We are looking to discussing with you the questions that are addressed unsatisfactorily.
>
> Best,
>
> Authors of Paper ID 465

---

> ### Author Response · Authors · 2022-08-08
> **Sincerely Request Your New Comment**
>
> Dear Reviewer 8FQc,
>
> Thanks for your review again. As the deadline for the authors' response is approaching, we sincerely request your comment on our primary response. This will definitely give us a valuable chance to address the questions unsolved.
>
> Best,
>
> Authors of Paper ID 465

---

### Official Review · Reviewer_6dJj · 2022-07-12

**Rating:** 5
**Confidence:** 4
**Soundness:** 2 fair
**Presentation:** 2 fair
**Contribution:** 3 good

**Summary:**

The paper proposes a way to utilize external information for inpainting. To do this, the proposed method maintains a database of clustered features collected from the dataset. Then, for each region's encoded feature, a matching cluster is identified, and the features of that cluster is augmented to the encoded feature in a soft, differentiable way. The proposed method can be an independent add-on to existing inpainting methods, and it consistently improves performance on quantitative metrics that measure similarity with the ground truth, such as PSNR, SSIM, L1, or LPIPS.

**Questions:**

There is possibility that a baseline method was evaluated incorrectly. It seems that the paper preprocessed the CelebA dataset with facial landmark alignment and cropping (Figure 8 of Supp Mat). However, MISF seems to not have preprocessed the data (their results are on much larger crop around the face, and the faces are not necessarily upright). Therefore, taking the MISF model trained on the raw CelebA data, and evaluating it along with a model trained specifically on aligned images is unfair. Likely because of this, the MISF results of the paper are often blurry (Fig4(a) and Fig8 of Supp Mat), while the results in the original MISF paper not not blurry. It is also wrong to copy the MISF numbers from the original paper (i.e. MISF line with 34.494, 26.635, 21.553, ... in Table 4), and comparing it with the CICM results, because they were in fact run on different datasets (unaligned vs aligned&cropped faces). If MISF were trained and test on the same preprocessing, it may get better results.

Why is MISF number different between Table 3 and 4? In Table 3, CelebA PSNR is 34.302, 26.387, 21.289, ... In Table 4, it is 34.494, 26.635, 21.553.

How would compare the quality of this result with ComodGAN (Zhao et al., ICLR2021)? Quantitative comparison is not provided, but ComodGAN seems very nice in qualitative results.

How is the convolutional encoder (which predicts F_m) trained? If it is trained end-to-end with the losses of Eq9, how are the features C of the cross-image feature sets are updated accordingly? Every update of the network parameter would make the feature Cs outdated.

I am still not sure why Eq2, 3 and 4 are formulated in the particular form presented. There seems to be connection to online k-means clustering.

In Eq7, how is the corruption ratio estimated? Is it predicted by the network?

**Limitations:**

The authors does address the limitations.

**Strengths And Weaknesses:**

Strength

I find it interesting to utilize external features for better image inpainting.
The ablations are thoughtful, including single-image vs cross-image context, and different ways to create the bank of features.
The paper achieves consistently good results than the baseline methods. In particular, it can be plugged into multiple existing methods and improve on all of them.

Weakness

The exposition is a bit difficult to follow. In general, the paper focuses on how the method is formulated, rather than why. Regarding this, I have a few questions. Please see the Questions section.

All evaluation metrics measure how much the output matches the ground truth, but it may not directly correlated with realism. For example, since image inpainting is a multi-modal problem with diverse possible outputs, to minimize the L1 loss, the output needs to be at the median of all possible values. In the end, L1 score gives advantage to the outputs that are smooth and low saturation. For example, a pix2pix model trained only on L1 objective (https://phillipi.github.io/pix2pix/images/index_facades2_loss_variations.html) looks unrealistic, even though it does achieve good L1 loss. To separately evaluate realism, metrics like FID are used (ComodGAN, Zhao et al, ICLR2021).

---

> ### Author Response · Authors · 2022-08-01
> **Response to Reviewer 6dJj**
>
> **1. Use FID as a metric.**
>
> We have added the results of different methods in terms of FID to the revised paper.
>
> **2. The baseline method is evaluated incorrectly.**
>
> Thanks for pointing out this error. We re-train and re-evaluate RFR, JPG, and MISF on CelebA. In Table 4 of the revised paper, we update the performances of RFR, JPG, and MISF on CelebA. CICM still achieves better results.
>
> **3. MISF numbers are different in Tables 3 and 4.**
>
> Different methods (e.g., RFR, JPG, and MISF), which are presented in original papers, compute the convolution feature maps with different resolutions for inpainting. To control the impact of resolution on the performance in the ablation study, all methods in Table 3 use the feature map with the lowest resolution (see lines 247-249 of the paper). In Table 4, we keep their original settings for state-of-the-art comparison.
>
> **4. Comparison with ComodGAN.**
>
> CICM is a component with the generalized cross-image context for assisting different inpainting networks, rather than a stand-alone model that surpasses other methods. The generalization power of CICM has been justified in Section 5.2 “Combination with Different Inpainting Networks”. We compare the performances of the latest networks (RFR. JPG, and MISF) with/without CICM. With CICM, these networks achieve consistent improvements on Places2 and CelebA datasets. We also compare ComodGAN with/without CICM in the table below. With CICM,  ComodGAN achieves a better performance.
>
> ||||||||Places2|||||||||
> |:-:|:-:|:-:|:-:|:-:|:-:|:-:|:-:|:-:|:-:|:-:|:-:|:-:|:-:|:-:|:-:|
> |||PSNR↑|||SSIM↑|||L1↓|||LPIPS↓|||FID↓||
> |methods|0-20%|20-40%|40-60%|0-20%|20-40%|40-60%|0-20%|20-40%|40-60%|0-20%|20-40%|40-60%|0-20%|20-40%|40-60%|
> |Comod|31.24|24.46|20.26|0.952|0.843|0.696|0.768|2.411|5.015|0.0428|0.1308|0.2275|17.53|34.57|65.11|
> |Comod-CICM|31.36|25.42|21.87|0.960|0.857|0.712|0.730|2.281|4.152|0.0404|0.1154|0.1913|16.30|31.17|58.21|
>
> ||||||||CelebA|||||||||
> |:-:|:-:|:-:|:-:|:-:|:-:|:-:|:-:|:-:|:-:|:-:|:-:|:-:|:-:|:-:|:-:|
> |||PSNR↑|||SSIM↑|||L1↓|||LPIPS↓|||FID↓||
> |methods|0-20%|20-40%|40-60%|0-20%|20-40%|40-60%|0-20%|20-40%|40-60%|0-20%|20-40%|40-60%|0-20%|20-40%|40-60%|
> |Comod|34.23|26.67|21.48|0.968|0.889|0.770|0.485|1.614|3.815|0.0317|0.0943|0.1878|5.738|21.74|58.26|
> |Comod-CICM|34.34|27.68|22.78|0.973|0.898|0.794|0.454|1.476|3.326|0.0293|0.0824|0.1653|4.558|18.37|48.32|
>
> **5. How is the encoder trained? How are the cross-image features updated? Every update of the network parameter makes the features outdated.**
>
> We train the encoder end-to-end with the losses in Eq. 9. First, we update the network parameters in backward propagation. Next, we rely on the updated parameters to compute the new regional features, which are used to update the cross-image features in CICM.
>
> Our strategy of updating the cross-image features can reduce the outdated cross-image features. Given a regional feature computed by the latest network parameters, we update all cross-image features in the corresponding set, where the intra-set similarities are enhanced. Compared to the well-updated feature sets, the outdated feature sets lead to a larger loss of intra-set similarities, thus driving new regional features to be injected into the outdated feature sets. In Table 1, we have experimented with/without using the intra-similarity for training. The results have demonstrated the effectiveness of the intra-similarity.
>
> **6. Eqs. 2, 3, and 4 seem to be online k-means clustering.**
>
> Online k-means focuses on how to continuously update the cluster centers with new samples. Eqs. 2-4 share a similar spirit with online k-means at this point but represent a specific process of utilizing and updating the cross-image features for inpainting.
>
> $[Feature$ $utilizing]$
>
> The k-means clustering only stores the centers for different clusters. These centers can be regarded as the anchor features of CICM, which are associated with different sets of cross-image features. But the centers alone lack visual information for inpainting, as evidenced by the experimental results in Table 2, “anchor only”. In contrast, we use CICM to store the cross-image features, which are computed and used to update the anchor features (see Eqs. 2 and 3), to provide a richer context for augmenting the regional features (see Eq. 4).
>
> $[Feature$ $updating]$
>
> Online k-means relies on the new samples to update the cluster centers. It assumes that all samples contain contemporary information. In our work, we use Eq. 2 to use the new regional features to update the cross-image features in different sets. Moreover, we focus on reducing the outdated cross-image features, by using the inter- and intra-set similarities to drive the feature updating. It refreshes the complex cross-image features, which are learned from the images with diverse contents.
>
> **7. How are the corruption ratios estimated?**
>
> The ratios are estimated by the network.

---

> ### Author Response · Authors · 2022-08-07
> **Response to Reviewer 6dJj**
>
> Dear Reviewer 6dJj,
>
> We thank you again for your valuable comments, which significantly help us to polish our paper. We are looking to discussing with you the questions that are addressed unsatisfactorily.
>
> Best,
>
> Authors of Paper ID 465

---

> ### Author Response · Authors · 2022-08-08
> **Sincerely Request Your New Comment**
>
> Dear Reviewer 6dJj,
>
> Thanks for your review again. As the deadline for the authors' response is approaching, we sincerely request your comment on our primary response. This will definitely give us a valuable chance to address the questions unsolved.
>
> Best,
>
> Authors of Paper ID 465

---

### Official Review · Reviewer_Kgu7 · 2022-07-12

**Rating:** 5
**Confidence:** 4
**Soundness:** 3 good
**Presentation:** 3 good
**Contribution:** 2 fair

**Summary:**

This paper proposes a Cross-Image Context Memory (CICM) for learning and using the cross-image context to recover the corrupted regions. It tries to provide richer context that benefits the inpainting task. The experimental results demonstrate the effectiveness and generalization of CICM.

**Questions:**

1. Please show some visualization evidence of different/identical set(s).
2. Figure 1 can not show that cross-image feature sets are useful for inpainting. Please provide more clear examples.

**Ethics Review Area:**

["Privacy and Security (e.g., consent)"]

**Limitations:**

1. Using external memory can improve the performance, also it will bring a lot of computational costs. Please make some analysis.


**Strengths And Weaknesses:**

Strengths:
1. The idea of using external memory is reasonable.
2. The paper is clearly written.

Weaknesses:
1. The method is a little common.
2. The advantages of external memory need more discussion.
3. The framework (Figure 2) is a little hard to follow. (c) and (d) provide the details but they are not necessary.
4. The aim of maximizing/minimizing the inter-/intra-set similarities between the cross-image features is not clear. Why is it useful for impainting?

---

> ### Author Response · Authors · 2022-08-01
> **Response to Reviewer Kgu7**
>
> **1. The method is a little common. What are the advantages of external memory?**
>
> Thanks for your useful comment. It helps us to clarify the advantage of our method.
>
> $[The$ $advantage$ $of$ $using$ $the$ $external$ $memory$ $for$ $image$ $inpainting]$
>
> Image inpainting relies on the context of the relevant regions for recovering the corrupted regions. The existing methods propagate the context of the surrounding regions to recover the corrupted regions in the same image. They learn the single-image context but yield unsatisfactory performances when the corrupted images lack information. The external memory stores the cross-image context learned from different images. Thus, it takes the advantage of more useful regions across images, providing a richer context for recovering a region.
>
> $[The$ $advantage$ $of$ $using$ $CICM$ $for$ $image$ $inpainting]$
>
> Our method of learning the cross-image context in the external CICM is non-trivial. The existing methods use the single-image context or class-specific information to recover the corrupted images. In contrast, we construct CICM, where the cross-image context is learned from rich image data. Moreover, CICM allows the inpainting to benefit from not only a kind of specific context. As evidenced in Section 5.2 “Extensive Evaluation on Semantic Inpainting” of the paper, CICM can store various kinds of cross-image context, which are learned from RGB images and semantic segmentation results.
>
> $[Clarification$ $of$ $the$ $major$ $contribution$ $of$ $this$ $paper]$
>
> Our major contribution not only lies in the methodology ground but also in the extensive thinking, evaluation, and discussion of CICM. CICM is a separable component alongside the inpainting network. Is it possible to train CICM with an network and apply it to another network? Is it also possible to train CICM on a dataset and evaluate it on another dataset? These questions are answered by very few works. In Section 3.3 of the supplementary file, we answer the above questions by evaluating CICM in the cross-model and -dataset scenarios. The cross-model CICM achieves better results than the methods without CICM. It demonstrates the capacity of CICM for transferring the context between different models. Yet, CICM degrades the performances in the cross-dataset scenario, demonstrating its limitation. We have provided our explanation of the performance degradation, pointing out a direction for improving the generalization of CICM in the future.
>
> **2. The framework (Figure 2) is a little hard to follow. (c) and (d) provide the details but they are not necessary.**
>
> We have revised Figure 2(c-d), by trimming the redundant arrows and re-organizing the legends. Please see the revised paper.
>
> **3. The aim of maximizing/minimizing the inter-/intra-set similarities between the cross-image features is not clear. Why is it useful for inpainting?**
>
> We are sorry for reversing the terms of inter- and intra-set similarities in Eq. 9. Actually, we minimize the inter-set similarity, for enhancing the diversity of the cross-image features in different sets. We maximize the intra-set similarity to encourage the cross-image features in the identical set to contain consistent context. We have updated Eq. 9 and its description in the revised paper (lines 151-155). We have updated Figure 5 in the supplementary file, where the inter- and intra-set similarities are reversed. We have double-checked our implementation, making sure the losses are implemented correctly. The diversity in different feature sets provides more chances to find the useful context from CICM for inpainting. The consistency in the identical set reduces the unreasonable cases, where discrepant contents are predicted for similar regions. Their effectiveness has been evaluated in Table 1 of the paper.
>
> **4. Visualization evidence of different/identical set(s).**
>
> In Figure 6 of the supplementary file, we have visualized the distribution of the cross-image features in different/identical set(s). We use t-SNE to map these features into a 2D latent space for visualization. Different/identical set(s) of the cross-image features appear closely/far. It means that they have a large diversity/consistency.
>
>
> **5. Figure 1 cannot show that cross-image feature sets are useful for inpainting.**
>
> We have replaced the example in Figure 1 in the revised paper. In the new example, the cars are required to be recovered. However, a large portion of the image is corrupted, thus lacking visual information about cars. By using the cross-image features in CICM (see the feature set 1), we find the relevant context of cars for recovering the image.
>
> **6. External memory brings a lot of computational costs.**
>
> We have discussed this limitation in Section 3.2 of the supplementary file. We compare the network parameters, GPU memory, and FLOPs of the networks with/without CICM, along with the performances. One may use these results to consider the trade-off between performance and efficiency.

---

> ### Author Response · Authors · 2022-08-07
> **Response to Reviewer Kgu7**
>
> Dear Reviewer Kgu7,
>
> We thank you again for your valuable comments, which significantly help us to polish our paper. We are looking to discussing with you the questions that are addressed unsatisfactorily.
>
> Best,
>
> Authors of Paper ID 465

---

> ### Author Response · Authors · 2022-08-08
> **Sincerely Request Your New Comment**
>
> Dear Reviewer Kgu7,
>
> Thanks for your review again. As the deadline for the authors' response is approaching, we sincerely request your comment on our primary response. This will definitely give us a valuable chance to address the questions unsolved.
>
> Best,
>
> Authors of Paper ID 465

---

### Author Response · Authors · 2022-08-01
**Deep gratitude to all reviewers for their valuable comments.**

We express our deep gratitude to all reviewers for their valuable comments, which significantly help us to better clarify our technical contributions and evaluate the effectiveness of our method. Below, we provide our point-to-point responses to the questions raised by all reviewers.

---

### Meta-Review · Area_Chair_FJcL · 2022-08-25

**Recommendation:** Accept
**Confidence:** Certain

**Metareview:**

The paper discusses how to use external information for inpainting. Reviewers appreciated the idea but raised concerns regarding limited novelty, use of the proposed method for inpainting, baselines being evaluated incorrectly, and missing ablations. The rebuttal was able to address most of the concerns and reviewers remained positive. AC concurs and doesn't find reasons to overturn an unanimous majority recommendation.

**Award:**

No

---

### Decision · Program_Chairs · 2022-09-14

Accept